# A 430 kyr record of ice-sheet dynamics and organic-carbon burial in the central Eurasian Arctic Ocean

Ruediger Stein [1,2,3] ✉, Thomas Frederichs [1], Kirsten Fahl [2], Walter Geibert [2], Jens Matthiessen [2], Frank Niessen[2], Christoph Vogt [1], Cynthia Sassenroth[2,4] & Evgenia Bazhenova [1]

Late Quaternary central Arctic Ocean paleoclimatic records going beyond the last climate cycle, are still very rare. Here, we present data from a well-dated deep-sea sediment core that allow for the first time to reconstruct in detail the interrelationship between ice-sheet dynamics and organic-carbon (OC) burial in the Eurasian Basin during the last 430 kyr, and to correlate marine and terrestrial records of the Eurasian Ice Sheet (EIS) history. Using a multi-proxy approach, we demonstrate that within pulses of EIS advance and retreat during glacial to subsequent deglacial times, erosion of ancient (petrogenic) OC-rich sedimentary rocks and soils deliver sediments, further supplied onto the shelf and beyond. Down-slope and long-distance transport of the fine-grained OC-rich suspension by turbidity and contour currents cause elevated burial of terrestrial OC and anoxic sedimentary conditions in the deep Eurasian Basin that allow preservation of labile algae-type OC.

Within the ongoing debate of recent climate change, the remote Arctic Ocean with its strong seasonal forcing and variability in runoff, sea ice formation, and sunlight, became a major focus during the last few decades as the high latitudes mirror the global warming trend strongly amplified[1,2]. Due to complex feedback processes (collectively known as "polar amplification"), the Arctic is both a contributor of climate change and a region that will be most affected by global warming. That means, the Arctic Ocean and surrounding areas are (in real time) and were (over historic and geologic time scales) subject to rapid and dramatic change. On geological time scales, the glacial history of the Arctic Ocean characterized by the repeated waxing and waning of circum-Arctic continental ice sheets (Fig. 1 showing the ice-sheet extent during MIS 6), the growth and disintegration of ice shelves, and related changes in ocean circulation patterns and sea-ice cover[3–8], have to be taken into account as well. There is, however, still an ongoing and partly controversial debate about the timing and extent of maximum glaciations and related interrelationships with changes in climate and

environment on land and in the ocean. A key problem in this debate about the history and significance of the Arctic cryosphere within the climate system, that has not been fully resolved yet, is the accurate dating of the Quaternary sediment sequences, a major challenge for the entire Arctic research community for several decades.

When talking about the (global) climate system and its driving processes, the organic carbon (OC) cycle plays a key role, i.e., production on land and in the oceans, storage/burial in the ocean and sediments, diagenesis/degradation, and release of greenhouse gases to the atmosphere[9]. Concerning the modern Arctic Ocean OC cycle, i.e., OC origin and burial in the sediments, there is a clear predominance of terrestrially-derived OC, and huge amounts of terrestrial OC (TerrOC) are currently buried in circum-Arctic shelf sediments, soils and peat deposits and stored frozen in permafrost soils and ice complex deposits[10–12]. As a result of increased surface warming in Arctic regions, a widespread decay of permafrost, collapse and erosion of coastal ice complex deposits, and the decomposition of hydrates

[1]Faculty of Geosciences and Center for Marine Environmental Sciences (MARUM), University of Bremen, Bremen, Germany. [2]Alfred Wegener Institute (AWI) Helmholtz Centre for Polar and Marine Research, Bremerhaven, Germany. [3]Frontiers Science Center for Deep Ocean Multispheres and Earth System and Key Laboratory of Marine Chemistry Theory and Technology, Ocean University of China (OUC), Qingdao, China. [4]Present address: Department of Earth Sciences, University of Gothenburg, Göteborg, Sweden. ✉ e-mail: ru_st@uni-bremen.de

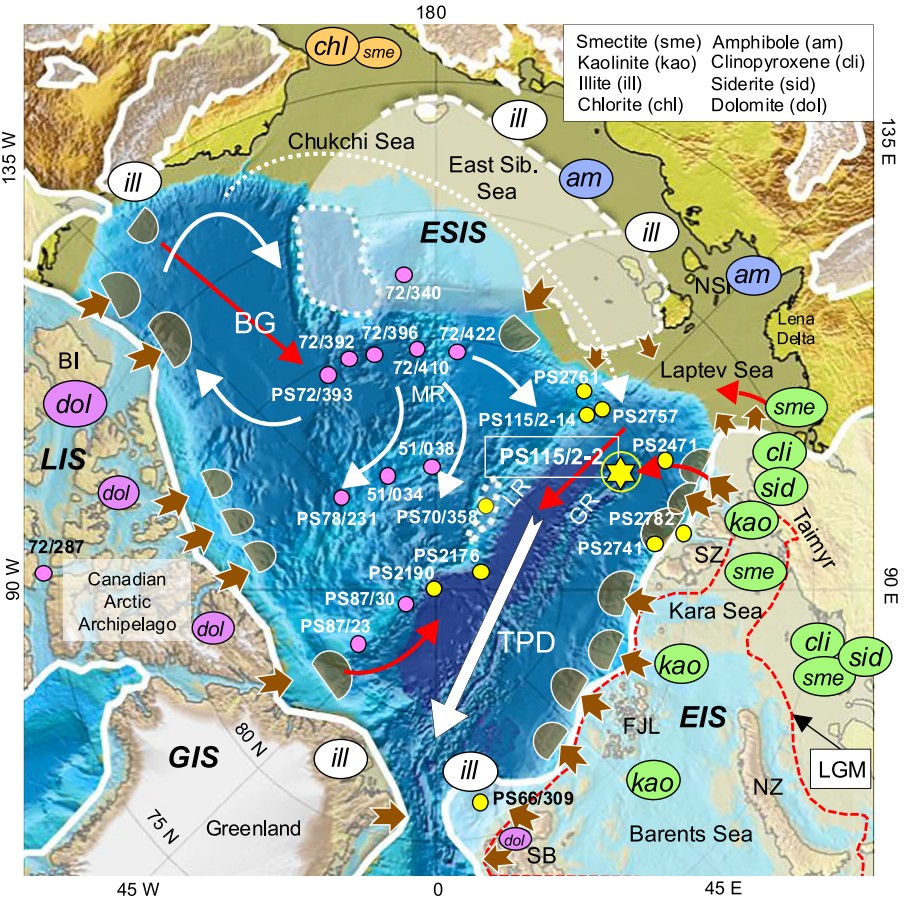

**Fig. 1 | Bathymetric map showing maximum circum-Arctic Pleistocene (MIS 6) glaciations[4] and mineral assemblages indicative of specific source areas (ref. 56, updated and references therein).** Location of Core PS115/2-2 is shown as yellow asterisk, locations of other cores mentioned in the text as yellow circles. Furthermore, the pink circles mark core locations with major dolomite-rich coarse-grained ice-rafted debris (IRD) from Canada (see Supplementary Figs. 16 and 17). Brown arrows indicate ice-sheet erosion and mass-wasting sediment input, with glacial-sedimentary depocenters (trough-mouth fans) in front of the major cross-shelf troughs (highlighted as brownish-transparent half-cycles)[24,102]. Red arrows indicate turbidity currents, white arrows the major surface-water current systems (BG Beaufort Gyre and TPD Transpolar Drift). White stippled arrow indicates an extended Beaufort Gyre situation under anticyclonic conditions typical for negative phase of Arctic Oscillation[86] that would allow the transport of detrital sediments from the Canadian Arctic into the Eurasian Basin. Stippled red line indicate extent of the EIS during the last glacial maximum (LGM). GR Gakkel Ridge, LR Lomonosov Ridge, MR Mendeleev Ridge, FJL Franz Josef Land, NSI New Siberian Islands, NZ Novaya Zemlya, SB Svalbard, SZ Severnaya Zemlya, GIS Greenland ice sheet, LIS Laurentide ice sheet, ESIS East Siberian ice sheet, EIS Eurasian ice sheet, BG Beaufort Gyre, TPD transpolar drift. Color codes mark source region: green = western Laptev Sea and Kara Sea, Barents Sea; blue = eastern Laptev Sea, East Siberian Sea; orange = Bering Strait; pink = Canada, northern Greenland; white = no specific source area.

could lead to the release of large quantities of $CH_4$ (and $CO_2$) to the atmosphere as well as an enhanced mobilization and export of ancient, previously frozen soil-derived OC[13–16].

Looking at geological time scales, the last deglacial atmospheric $CO_2$ increase may be explained by the release and subsequent oxidation of large amounts of ancient TerrOC[17,18]. On the other hand, during glacial climate cooling and times of extended circum-Arctic ice sheets and lowered sea-level, huge amounts of OC-rich soils, sediments and rocks might be eroded and transported by glacigenic processes across the shelf edge, and buried in deep ocean sediments, thus acting as an important geological $CO_2$ sink and storage (cf., refs. 19,20). Well-dated detailed records dealing with such glacigenic and oceanographic processes, sediment discharge, and rates of OC burial in the central Arctic Ocean during late Quaternary glacial stages, however, are quite limited and mainly restricted to the last climate cycle.

Here, we have utilized a sediment core recovered from the Amundsen Basin in 3600 m water depth at the eastern flank of the Gakkel Ridge during Polarstern Expedition PS115/2 in 2018 (ref. 21),

representing four glacial/interglacial cycles of the last 430 kyr. Our age model, a fundamental prerequisite for any kind of paleoclimatic reconstruction, is based on records of the inclination and relative paleointensity (RPI) of the Earth's magnetic field approved by rock magnetic data, $^{230}Th$ and $^{231}Pa$ excess ($^{230}Th_{ex}$ and $^{231}Pa_{ex}$ records, $^{230}Th_{ex}$ maxima, and AMS$^{14}$C ages. Using organic-geochemical proxies, we have identified prominent well-defined sections with strongly elevated concentration of ancient (petrogenic) predominantly terrestrial OC, coinciding with glacial time intervals of extended ice sheets and the subsequent terminations/deglacials. Mineral assemblages related to the complex geology of the surrounding hinterland (Fig. 1), in combination with sediment structures and textures are used to identify the provenance and transport pathways of these OC-rich sediments. Our multi-proxy approach allowed for the first time a detailed reconstruction of the OC burial history and its relationship to the Eurasian ice sheet (EIS) history, sediment and meltwater discharge, and depositional environment during the last about 430 kyr. Based on the preservation of labile algae-type OC under anoxic sedimentary

conditions, we demonstrate that even during strong glacial intervals there must have been at least occasionally open-water (polynya-type) conditions along the Eurasian continental margin in front of the ice sheet with marine and sea-ice algae productivity. Furthermore, we show that the glacial rates of predominantly TerrOC burial are three to six times higher than those determined in Holocene Arctic deep-sea sediments representing interglacial conditions, suggesting that Arctic deep-sea sediments (and especially those in the Eurasian Basin) were an important sink for TerrOC during glacial intervals. Finally, the prominent dolomite-rich pink layer ("Pink-White Layer 2") dated to about 282 ka, represents a key marker horizon to be used for ocean-wide correlation and dating of Arctic sediment cores. In summary, we are confident that Core PS115/2-2 is a key core for reconstructing Arctic paleo-climate conditions that is of major interest for the broader Arctic scientific community and beyond (in this context see also Supplementary Information Nos. 1–7 for additional important data).

## Results

### Sediment characteristics and lithostratigraphic framework
Polarstern Expedition PS115/2 was carried out in 2018 in the area of the southern Lomonosov Ridge and the Amundsen Basin towards the Laptev Sea continental margin (see Supplementary Information No.1). The expedition's program was mainly devoted (1) to finalize the exact drilling locations for the IODP Expedition 377 "Arctic Ocean Paleoceanography" (scheduled for August/September 2022 but unfortunately canceled in February 2022 due to political reasons) and (2) to reconstruct in detail the short- and long-term changes in circum-Arctic ice sheets, sea-ice cover, and surface- and deep-water characteristics[21]. In order to reach the latter goals, a Hydrosweep and Parasound survey was carried-out to select optimum locations for obtaining long undisturbed sediment cores (Supplementary Fig. 1). At the eastern flank of the Gakkel Ridge representing well-stratified pelagic sediments (Supplementary Fig. 1b, c), a complete and undisturbed composite sedimentary sequence was recovered using a kastenlot corer (Supplementary Fig. 2; for details see "Methods" below). This 764 cm long sequence of Core PS115/2-2 is characterized by two main lithologies: (1) brown to dark brown and light olive brown, partly bioturbated silty clays with alternations of thin silty clay/silt/sandy layers of varying brownish/grayish colors, and (2) very prominent distinctive intervals of dark gray silty clays (Supplementary Figs. 2 and 3)[21].

Such dark gray intervals that are enriched in OC (see below for details) have also been found in several sediment cores from the area across the southern Lomonosov Ridge (Supplementary Fig. 4) and along the continental margin of the Laptev and East Siberian seas and assigned to glacial intervals[21–25]. In most of these sediment cores, only one or two intervals with dark gray sediments have been recovered in the lowermost part of the sedimentary sections, and often the issue of accurately dating these cores has not been resolved yet (Supplementary Fig. 4)[7,25,26]. Furthermore, in cores from the NW-Svalbard, Eurasian and Siberian continental margins located more proximal to the glacial ice sheets, the dark gray sediments often represent coarser-grained ice-rafted debris (IRD)-rich or even diamicton facies, related to extensive glacial activities at times of extended ice sheets on the shelf and beyond[22,24,27,28]. In contrast, Core PS115/2-2 contains seven intervals of dark gray very fine-grained (no IRD) silty clays that are characterized by a sharp base and display prominent internal fining-upward structures, interpreted as distal turbidites and/or deposition from contour currents (Supplementary Fig. 3). Such a lithostratigraphic sequence with a series of "cyclic" events represent significant recurrent changes of the depositional environment including sedimentary processes, detrital sediment input, and OC burial rates, i.e., changes that are highly probable related to glacial/interglacial climate variability. Before any paleoenvironmental reconstruction can be done, however, the lithostratigraphic framework has first to be transferred into a reliable chronostratigraphic framework— still a major challenge for

many decades when working on Arctic Ocean sedimentary sections representing time intervals beyond the use of AMS[14]C dating[29–32]. Just based on shipboard data and the assumption that the dark gray intervals represent glacial intervals[21,23], a preliminary age model had been proposed, suggesting that the sedimentary sequence of Core PS115/2-2 might represent the last about 500 kyr (ref. 21). However, robust chronostratigraphic tie points as established in our study here, were missing at that time.

### Establishing an accurate chronology—a challenging task in Arctic paleoclimate research
Whereas dating techniques such as oxygen isotope stratigraphy, magnetostratigraphy, and biostratigraphy are routine tools in studies from most world ocean areas, several problems are obvious for Arctic Ocean sediments resulting in difficulties in establishing accurate age-depth relationships in the existing sediment cores. Poor preservation of calcareous and biosiliceous microfossil faunas and floras in Arctic cores often precludes application of conventional biostratigraphic and isotopic dating techniques. Paleomagnetic studies carried out on marine sediments from various regions of the Arctic Ocean, led to quite different age models and related sedimentation rates because the changes between reverse and normal polarities may be interpreted differently, i.e., as long-term reversals of the geomagnetic field, short-term geomagnetic excursions and/or a result of diagenetic overprint in the initial natural remanent magnetization (NRM)[33–39]. Because of these uncertainties in interpretation of magnetic inclination changes it is crucial to provide independent age control to guide the interpretation of the paleomagnetic polarity pattern.

Our age model for Core PS115/2-2 is based on a multi-proxy approach (Fig. 2; Supplementary Tables 1 and 2). We have used magnetostratigraphic proxies, i.e., inclination changes and RPI variations of the geomagnetic field, combined with rock magnetic measurements to assess a possible diagenetic overprint on the paleomagnetic signal. As independent age proxies, extinction ages of $^{230}$Th and $^{231}$Pa excesses and further $^{230}$Th$_{ex}$ maxima[31,40–42] as well as AMS[14]C ages have been determined.

The inclination record of Core PS115/2-2 is dominated by normal polarity (88% of the samples), interrupted by a series of prominent short intervals of reversed polarity (Fig. 2a). The RPI record of Core PS115/2-2 displays distinct cyclic variations with the prominent RPI minima coinciding with the geomagnetic excursions (Fig. 2a). When looking at the RPI and inclination records, however, it is obvious that in several instances RPI maxima coincide with the dark gray intervals (representing OC-rich, probably anoxic conditions; see below), and that the paleomagnetic reversals coincide mostly with beige-brownish (oxic) sediments (Fig. 2a).

Before using the inclination reversals and the RPI values as indicative for stratigraphy/chronology, diagenetic effects/overprint have to be excluded. Thus, to obtain information on the magnetic mineral composition and the degree of eventual oxidation, we carried out temperature-dependent measurements of magnetic remanence on selected samples characterized by both normal and reversed polarity and different OC-rich/OC-poor lithologies (Fig. 2a). The results and interpretation of these measurements are outlined in detail in the Supplementary Information No.2 and Supplementary Figs. 5–7. They allow the conclusion that there is no systematic difference in the magnetic properties between normal and reverse samples, and therefore that there is no obvious effect on the magnetostratigraphy for core PS115-2-2. Thus, our inclination record can be considered a reliable record of the Earth's magnetic field, reflecting intervals of normal and reverse polarities.

As an independent age control for supporting or rejecting the option of short-term geomagnetic excursions that have caused the negative inclinations, $^{230}$Th$_{ex}$ and $^{231}$Pa$_{ex}$ have been determined (see "Methods" for details). Both $^{230}$Th (with a half-life time of 75,690 years)

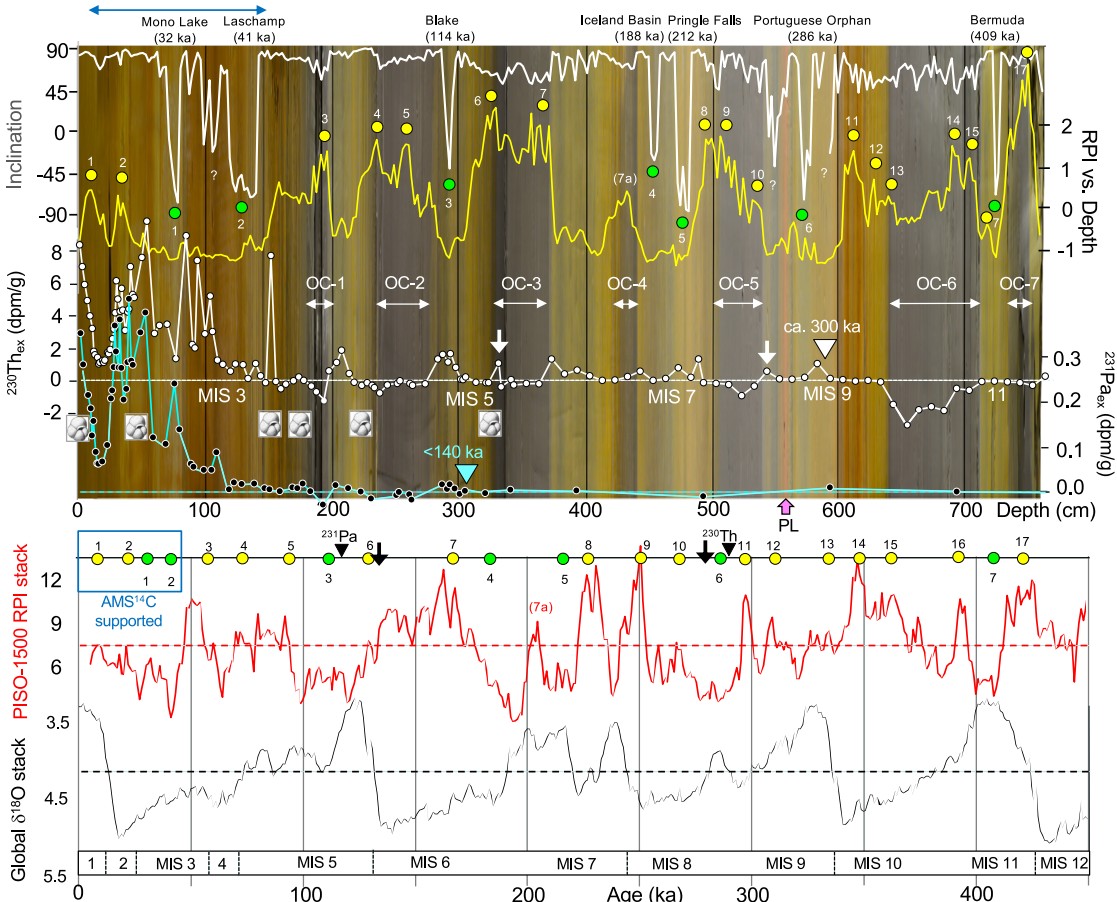

**Fig. 2 | Proxies for the development of the PS115/2-2 age model. a** Top to bottom: paleomagnetic inclination, relative paleointensity (RPI), excesses in $^{230}$Th ($^{230}$Th$_{ex}$) and $^{231}$Pa ($^{231}$Pa$_{ex}$), determined in Core PS115/2-2 and plotted versus core depth. At the top, our interpretation of the reversed inclinations as geomagnetic excursions with ages (according to ref. 38) are listed. Paleomagnetic tie points used for the development of the proposed age model (see Fig. 3) are indicated as yellow circles (RPI peaks correlated with the global PISO-1500 RPI stack of ref. 49) and green circles (negative inclinations assigned to geomagnetic excursions after ref. 38). The age model is further supported by AMS$^{14}$C dating (upper part of the sedimentary sequence marked by horizontal blue arrow, see Fig. 3 for details) and by $^{230}$Th$_{ex}$ (white triangle) and $^{231}$Pa$_{ex}$ (cyan triangle) extinction ages[31] as well as $^{230}$Th$_{ex}$ maxima correlated with the MIS 8/9 and MIS 6/5 boundaries (white arrows). The occurrence of planktic foraminifers is indicated by a foraminifer symbol. As background of the records, the emerged photograph of the PS115/2-2 sedimentary sequence (see Fig. 3) is shown, highlighting the dark gray organic-carbon-rich intervals OC-1 to OC-7. Pink arrow highlights the prominent pink layer (PL) (see Supplementary Fig. 17). **b** For comparison, the global benthic oxygen isotope stack[50] and global virtual axial dipole moment (VADM)-calibrated PISO-1500 RPI stack[49] are plotted vs. age. The age-model tie points of (**a**) are added at the top. Marine isotope stages (MIS) 1–12 are indicated. Source data of (**a**) are provided as a Source Data file.

and $^{231}$Pa (with a half-life time of 32,760 years) are particle-reactive radionuclides produced uniformly in the ocean by alpha decay of soluble $^{234}$U and $^{235}$U, respectively, and provide useful information for dating deep-sea sediments[43]. Assuming as a first approximation that $^{230}$Th$_{ex}$ and $^{231}$Pa$_{ex}$ remain detectable for four to five half-lives (around 300 kyr for $^{230}$Th$_{ex}$ and 140 kyr for $^{231}$Pa$_{ex}$)[31], this age limit is found at core depths of about 590 cm and 300 cm, respectively (Fig. 2). We use here an approach of using the last confirmed presence of water column-derived excess. This approach is complementary to the $^{230}$Th$_{ex}$ and $^{231}$Pa$_{ex}$ extinction age of Hillaire-Marcel et al. (ref. 31), but it relies on the last positive identification of $^{230}$Th$_{ex}$ or $^{231}$Pa$_{ex}$ rather than confirming its continuous absence, which might have other causes than decay only[44,45] and is more complex[26,42]. The sequence of significantly positive $^{231}$Pa$_{ex}$ measurements in a strictly oxic part of the sediment around 300 cm depth (Fig. 2a), marks this point as clearly post-MIS 6. Together with the distribution of $^{230}$Th$_{ex}$, these radiometric age constraints support our chronostratigraphic framework that the prominent negative inclinations at 573 and 293 cm represent the Portuguese Orphan and Blake geomagnetic excursions, respectively (Figs. 2 and 3). Taking together the $^{230}$Th$_{ex}$ and $^{231}$Pa$_{ex}$ data in combination with the

paleomagnetic records we use 290 ka (MIS 9a) and 120 ka (MIS 5) for the last confirmed presence of water-column derived $^{230}$Th$_{ex}$ and $^{231}$Pa$_{ex}$, respectively (Supplementary Fig. 8). Intervals with elevated $^{230}$Th$_{ex}$ values are correlated with interglacials (late MIS 9, MIS 7, MIS 5, MIS 3 and MIS 1). For MIS 5, MIS 3, and MIS 1, this correlation is also in line with the occurrence of planktic foraminifers (Fig. 2). The intercalated glacial (stadial) intervals, on the other hand, display minimum $^{230}$Th$_{ex}$ values < 0, which may be related to uranium mobility around oxygen-deficient parts of the sediment. Based on this $^{230}$Th$_{ex}$ record, the MIS 9/8, MIS 8/7 and MIS 6/5 boundaries are at about 544 cm, 500 cm and about 325-284 cm core depth, respectively, and the upper 150 cm are of MIS 3 to MIS 1 age (Fig. 2 and Supplementary Fig. 8). This stratigraphic age/depth assignment is very similar to that determined in a $^{230}$Th$_{ex}$ record of Core PS2757 recovered close-by from southern Lomonosov Ridge (Supplementary Fig. 8f; for core location see Fig. 1)[26], further supporting our chronostratigraphic approach.

For the upper 180 cm, additional support for our age model comes from AMS$^{14}$C measurements. Unfortunately, sufficient numbers of foraminifers for dating are only present in the upper five centimeters. Thus, only three AMS$^{14}$C datings of planktic foraminifers are

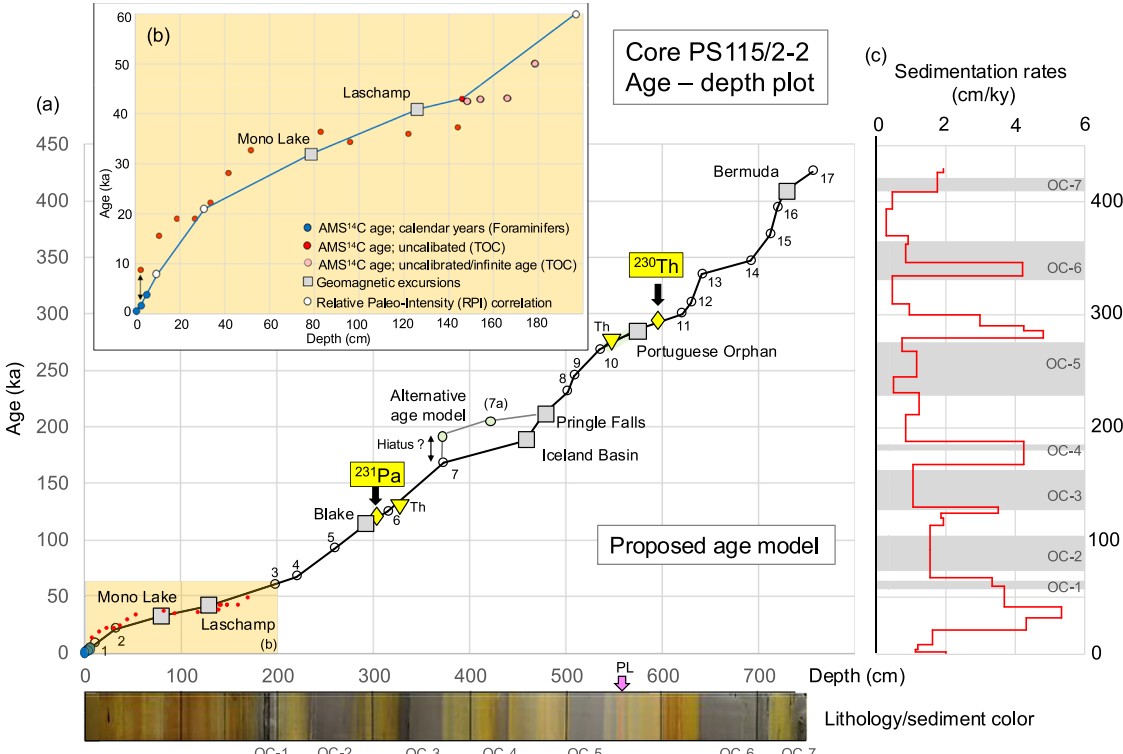

**Fig. 3 | Age-depth plot, proposed age model and sedimentation rates. a** Age-depth plot based on the tie points identified in Fig. 2. Blue and white circles and gray squares indicate data points based on AMS[14]C ages of planktic foraminifers, relative paleointensity (RPI) values (numbered 1–17), and geomagnetic excursions, respectively, and used for the development of our proposed age model (see Supplementary Table 2). Small red dots show uncalibrated AMS[14]C ages of acid-insoluble OC (AIOC) residues. In addition, [230]Th$_{ex}$ and [231]Pa$_{ex}$ extinction ages and two specific [230]Th$_{ex}$ peaks are added as yellow diamonds and yellow triangles, respectively (see text for further details). At the bottom depth scale, a photograph of the 764 cm thick sedimentary sequence of Core PS115/2-2 is added, showing color variations and sedimentary structures. The dark gray organic-carbon-rich intervals OC-1 to OC-7 are highlighted (for details see Supplementary Fig. 3). Pink

arrow highlights the prominent pink layer (PL) (see Supplementary Fig. 17). A minor hiatus at 373 cm core depth cannot be fully excluded (for details see Supplementary Information No.3 and Supplementary Fig. 8). **b** Blow-up of the uppermost 200 cm representing the last about 60 kyr to show samples for which AMS[14]C measurements have been carried-out. At 3 cm core depth, AMS[14]C ages from both planktic foraminifers and AIOC are available. The age difference between both values (highlighted as black arrow) is about 8–9 kyrs, indicating the presence of "old" reworked terrestrial OC. For further details see "Methods" and Supplementary Table 1. **c** Sedimentation rates (cm/kyr) based on our proposed age model and plotted versus age. The OC-1 to OC-7 intervals are highlighted as gray bars. Source data of (**a**) are provided as a Source Data file.

available that give ages between 0.2 and 3.2 ka (Fig. 3, Supplementary Table 1). These ages are calendar years before the present (for calculation see methods and Supplementary Table 1). Due to this lack of biogenic carbonate, we also used bulk acid-insoluble OC (AIOC) residues for AMS[14]C dating. The interpretation of such AMS[14]C dates, however, is more complicate as the samples may contain both fresh marine OC but also fresh and "old" (dead) reworked terrestrial OC[46–48]. Especially in Arctic Ocean sediments, the terrigenous OC fraction is mainly predominant[10]. Thus, the determined AIOC AMS[14]C ages may be several thousands of years older than the true age of deposition in the marine environment, i.e., the ages should represent the maximum ages of the marine sediments. This is in line with the AIOC-based AMS[14]C dates from Core PS115/2-2, indicating increasing ages with increasing depth, an offset of about 8 kyr between the AIOC and the foraminifera-based AMS[14]C dates at 3 cm core depth, and a maximum uncalibrated AIOC age of about 43 ka at 146 cm core depth (Fig. 3, Supplementary Table 1). Furthermore—and most important—these AIOC AMS[14]C dates strongly support that the prominent youngest negative inclinations at about 120–140 cm and 74–79 cm represent the Laschamp (41 ka) and Mono Lake (32 ka) geomagnetic excursions[38] (Fig. 3).

Our RPI record is compared with the global RPI/isotope stack "PISO-1500" that is based on coupled RPI and oxygen isotope records and developed as stratigraphic template for correlating and dating sedimentary records for the last 1.5 Myr[49] (for the last 450 kyr of the

stack see Fig. 2b). Both RPI records show a quite similar pattern, and alignment of the records is possible. Based on this correlation, RPI tie points for the establishment of our age model have been identified (Figs. 2 and 3). For an alternative interpretation of the RPI record we refer to the Supplementary Information No.3 and Supplementary Fig. 8.

Considering our independent proxy data sets we are quite confident that in our sediment core (study area), the negative magnetic inclinations represent short-term geomagnetic excursions within the normal Brunhes Chron and can be used as geochronological tie-points. We assigned the negative inclinations of Core PS115/2-2 to the Bermuda (409 ka), Portuguese Orphan (286 ka), Pringle Falls (212 ka), Iceland Basin (188 ka), Blake (114 ka), Laschamp (41 ka) and Mono Lake (32 ka) excursions (Fig. 2a)[38]. Neither the shallow inclinations at about 99–100 cm core depth nor the negative inclinations at about 550 and 594 cm below and above our Portuguese Orphan excursion, respectively, have been attributed to any excursion discussed in the literature. In conclusion, we are able to establish an accurate age model ("Age Model A"; Fig. 3 and Supplementary Fig. 8) that is based on seven geomagnetic excursion ages, 17 RPI tie-points, the two [230]Th$_{ex}$ and [231]Pa$_{ex}$ extinction ages, two [230]Th$_{ex}$ maxima as well as AMS[14]C dates for the uppermost part of the core (Figs. 2 and 3; Supplementary Tables 1 and 2). Plots of the magnetic inclination, RPI, and [230]Th$_{ex}$ records versus age are shown in Supplementary Fig. 8. According to

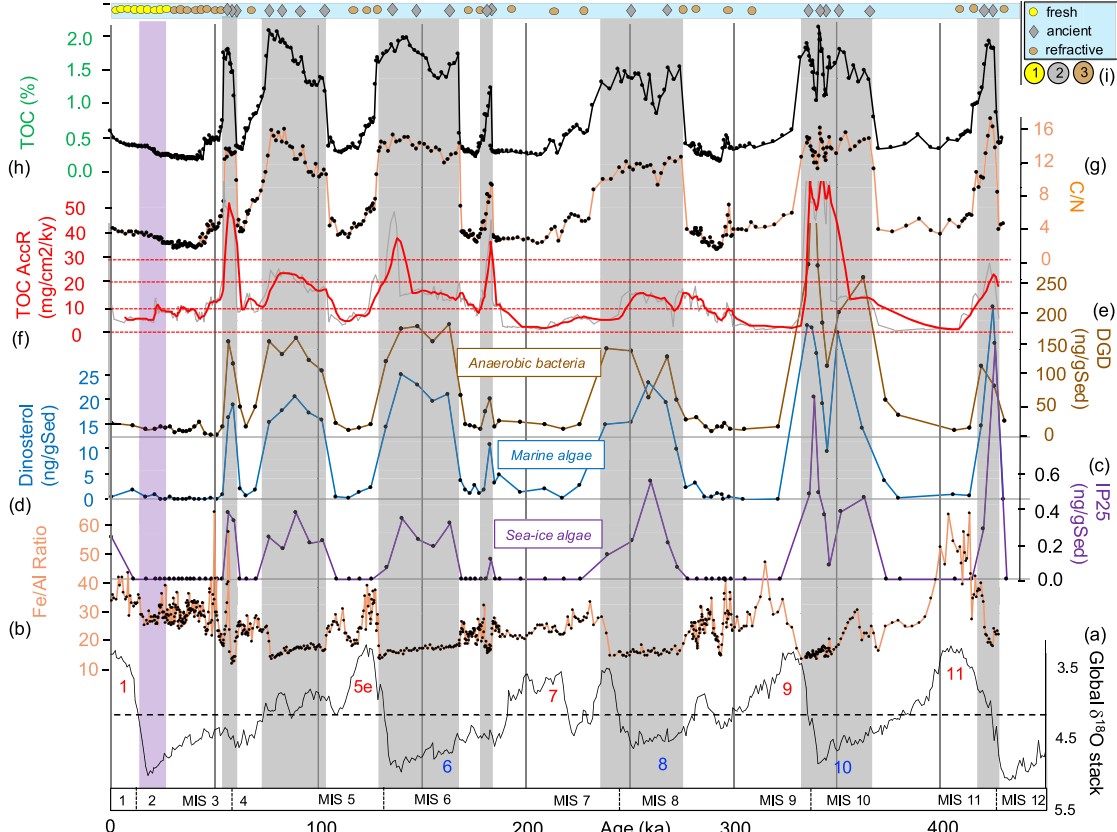

**Fig. 4 | Organic-geochemical proxy records for Core PS115/2−2 representing the last about 430 kyr. a** Global benthic oxygen isotope stack[50]. Glacial/interglacial cycles and MISs are highlighted. **b** Fe/Al record for Core PS115/2−2 (this study) that resembles quite well the isotope stack with maxima and minima coinciding with interglacials and glacials, respectively. **c**–**e** Selected biomarker concentrations (in ng/gSediment): IP25 as proxy for sea-ice algae productivity, dinosterol as proxy for open-water phytoplankton productivity, and $C_{33}$ dialkyl glycerol diether (DGD) as proxy for anaerobic bacteria (see "Methods" for further explanations) (this study). (**f**) Accumulation rates of total organic carbon (TOC in mg/cm²/kyr), in red three-point moving average (this study). **g**–**i** Organic-geochemical bulk parameters: C/N ratios calculated as TOC/TN ratios and indicative for the OC source; total organic carbon (TOC) content, and maturity of the OC based on Rock-Eval pyrolysis, distinguishing between fresh/immature OC, ancient/petrogenic OC and refractive/oxidized OC (for details see Fig. 5) (this study). The OC-1 to OC-7 intervals are highlighted as gray bars; the Last Glacial Maximum (LGM), characterized by OC-poor sediments, is highlighted by a purple bar. Source data of (**b**–**h**) are provided as a Source Data file.

this age model, the sedimentary sequence of Core PS115/2-2 represents four glacial/interglacial cycles (MIS 11 to Modern), i.e., the last 430 kyr. During this time span, sedimentation rates vary between 0.5 and 5 cm/kyr (Fig. 3c), reflecting distinct changes in climate-controlled environmental conditions.

## Glacial to interglacial variability in organic carbon sources and burial

Based on our age model and the comparison with the global benthic oxygen isotope stack[50], the prominent dark gray intervals characterized by elevated TOC values of 1–2% correlate with cold climate stages (Fig. 4a, h; Supplementary Table 3). In detail, OC-6, OC-5, and OC-3 intervals coincide with full glacial intervals (MIS 10, MIS 8, and MIS 6, respectively) and the subsequent terminations/deglacials. OC-4, a minor event with a TOC content of about 1%, falls into the early phase of MIS 6 (or a colder phase in the late MIS 7 if "Age Model B" is applied; see Supplementary Fig. 8), and OC-2 and OC-1 fall into late MIS 5 (MIS 5b?) and MIS 4/3. The age of the oldest OC-7 interval probably falls into the uppermost MIS 12/MIS 11 transition. The thicknesses of the OC-rich intervals are quite different. Whereas OC-6, OC-5, OC-3, and OC-2 have thicknesses between 36 and 68 cm, OC-4 and OC-1 (and OC-7?) have thicknesses of only 10–20 cm (Supplementary Table 3). Furthermore, it is important to note that during MIS 2, i.e., the Last Glacial Maximum (LGM), the sediments do not show increased TOC contents (Fig. 4h).

The brownish-beige sediments representing the intercalated interglacials, on the other hand, are characterized by low TOC values of 0.2–0.5% (Fig. 4h), i.e., they are quite similar to those determined in surface sediments from the central Arctic Ocean representing the modern interglacial conditions[10]. In terms of absolute OC burial, the glacial OC accumulation rates reaching about 15–30 mg cm⁻² kyr⁻¹ are about three to six times higher than those determined for interglacial time intervals (<5 mg cm⁻² kyr⁻¹) (Fig. 4f).

The elevated TOC contents in all seven OC intervals are predominantly of terrigenous origin (TerrOC) as reflected in the high C/N ratios of 10–16 (or >20 when corrected for inorganic nitrogen; see "Methods") (Fig. 4g; Supplementary Fig. 9), maxima in terrestrial biomarkers (campesterol, ß-sitosterol and long-chain n-alkanes; Supplementary Table 4) as well as the Rock-Eval pyrolysis data (Fig. 5). Furthermore, the Rock-Eval Tpeak (Tmax) data allow to classify the TerrOC in terms of its maturity stage (Fig. 5d; see "Methods" for further details). The upper about 40 cm section with Tmax values of 300–330 °C is characterized by immature TerrOC, although a contribution of some more mature ("older") OC is obvious from the bi-/polymodal signature of the pyrolysis temperature plots (Fig. 5e). Below 40 cm, two main OC types can be distinguished (Fig. 5): first, the brown-beige, OC-poor sediments that mainly contain highly degraded refractive/oxidized ("post-mature") OC with very low HC yield generated during pyrolysis (Fig. 5c); no Tpeak value was reached before the

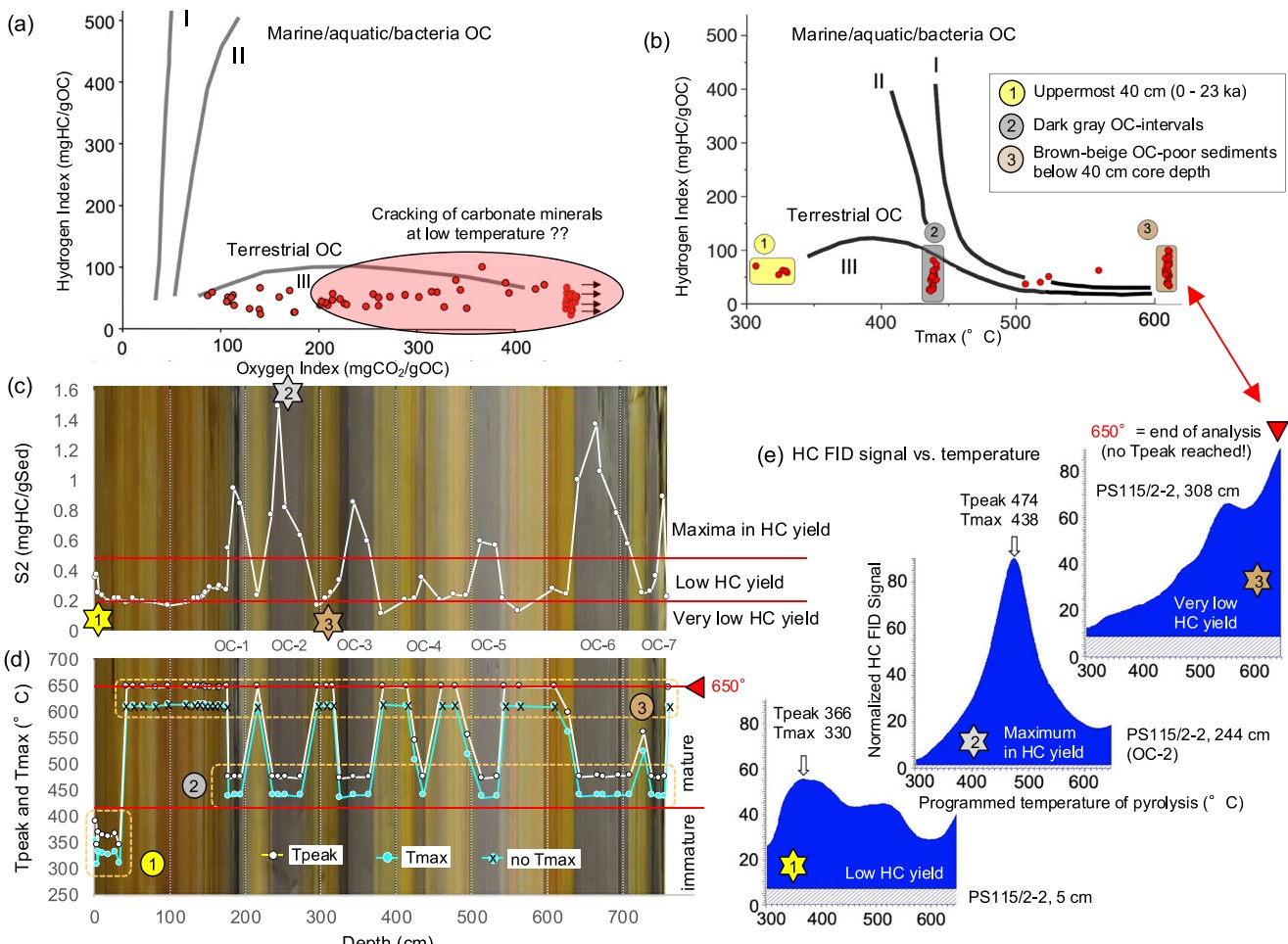

**Fig. 5 | Results of Rock-Eval pyrolysis of Core PS115/2−2 sediment samples (this study). a** Hydrogen vs. oxygen index ("van-Krevelan-type") diagram allowing to distinguish between kerogen types I, II, and III and indicating the predominance of terrestrial OC at Core PS115/2−2 (refs. 91,92). Based on the correlation between HI values and percentages of terrestrial/marine OC (derived from kerogen microscopy) and determined in Cenozoic Arctic Ocean sediments, a first-order estimate suggests that the terrestrial OC proportion is about >90% (refs. 10,103,104). The partly very high oxygen index values may suggest cracking of carbonate minerals (especially siderite) already at low temperature[94,95]. **b** Hydrogen index vs. Tmax diagram. Based on Tmax values, three groups of sediments with different stages of OC maturity have been identified: (1) fresh/immature OC, (2) mature (ancient/petrogenic) OC, and (3) highly degraded/oxidized OC. **c** Amount of hydrocarbons (HC) generated during pyrolysis between 300 and 650 °C heating interval (S2 in mg HC/gSed). Horizontal lines separate samples with very low, low, and maximum HC yields, respectively. Stars 1–3 mark the samples shown in (**e**). **d** Temperature of maximum (peak) generation of HCs during pyrolysis (Tpeak) and corrected value after apparatus vs. standard calibration (Tmax = Tpeak − 36°; cf. ref. 95). For the HC-lean samples of Group 3, no Tpeak value is reached before the pyrolysis end temperature of 650 °C, i.e., no Tmax value can be calculated. As background of the records, the emerged photograph of the PS115/2−2 sedimentary sequence (see Fig. 3) is shown, highlighting the dark gray OC-rich intervals OC-1 to OC-7. **e** Representative graphics for the groups 1–3, showing the measured HC flame ionization detector (FID) signal with increasing temperature during the pyrolysis. The three groups are characterized by very different HC yield; see (**c**). Large open arrows highlight Tmax values. Polymodal curves suggest the presence of different types of OC with different stage of maturity/degradation. Note: the HC signal numbers are normalized. Source data of (**a**–**d**) are provided as a Source Data file.

end of pyrolysis at 650 °C. Second, the dark gray OC-rich intervals with maximum HC yield generated during pyrolysis (Fig. 5c) that display a unimodal HC signature with distinct Tmax values of 435–440 °C, indicating mature (i.e., ancient/petrogenic) TerrOC (Figs. 4i and 5e). The presence of ancient/petrogenic carbon is further supported by the presence of sand-sized black particles, identified as coal in SEM photographs and EDAX analysis (Supplementary Fig. 9). These black particles are predominant in the >63 μm fraction but of minor significance when looking at the total abundance as the >63 μm fraction content in all OC intervals is <0.1% (Fig. 6b). However, the continental-derived coal might have been ground by glacigenic processes and mainly occurring in the silt/clay fraction.

Although TerrOC is clearly predominant in the OC-rich intervals, low but significant concentrations of more labile biomarkers dinosterol, brassicasterol and short-chain n-alkanes (indicative for open-water phytoplankton productivity), and specific highly-branched isoprenoids (IP25 and HBI-II indicative for sea-ice algae productivity) are preserved in the glacial sediments as well (Fig. 4c, d and Supplementary Table 4). On the other hand, such labile algae OC is absent or only present in very minor concentrations in the beige-brown sediments representing interglacial conditions. This difference in OC composition points to fundamentally different depositional environments during glacial and interglacial times (see "Discussion" for further details).

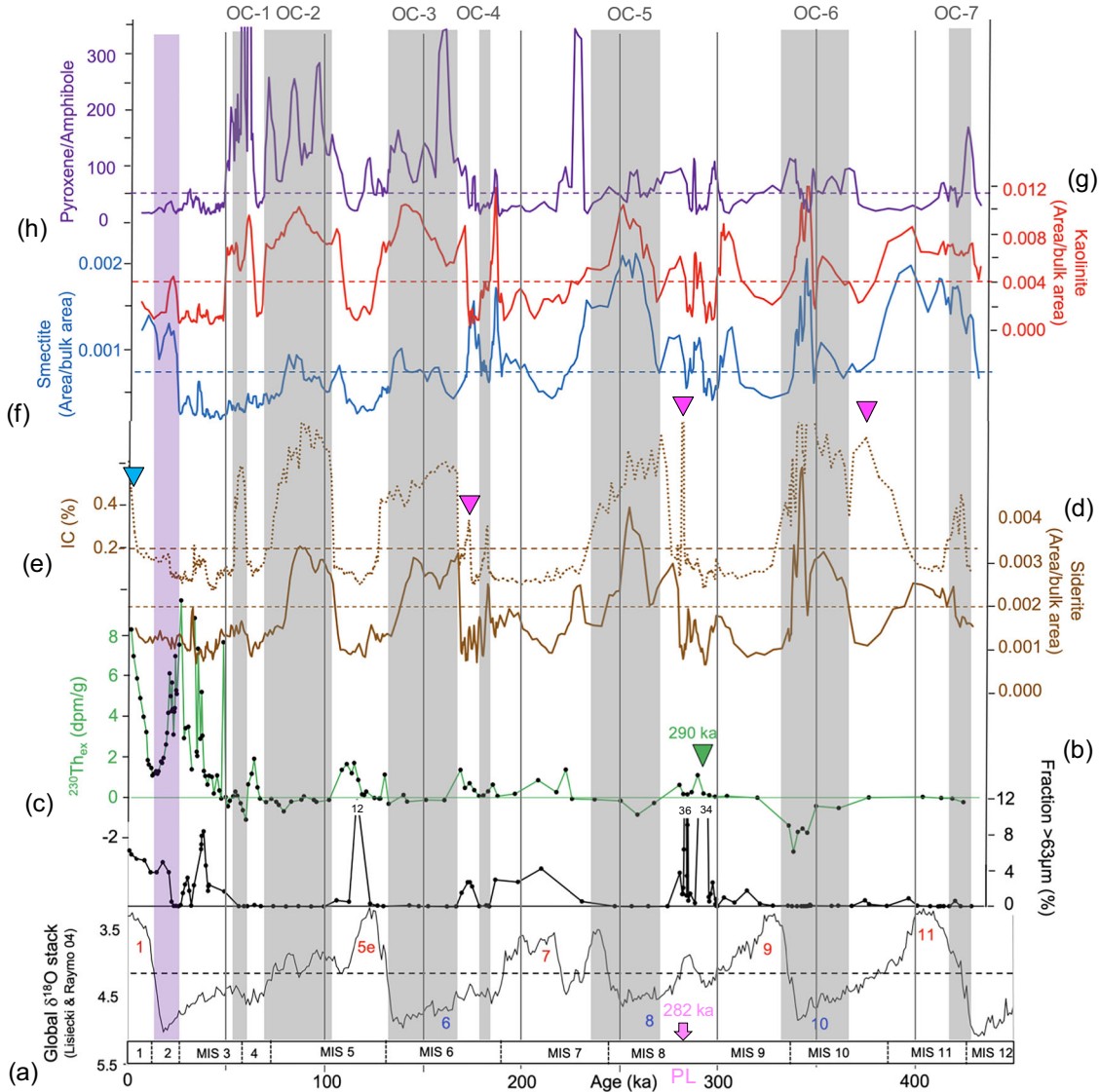

**Fig. 6 | Proxy records indicative for provenance and transport processes of detrital sediments of Core PS115/2−2, plotted versus age. a** Global benthic oxygen isotope stack[50]. Glacial/interglacial cycles and MISs are highlighted. **b** Content of coarse fraction >63 μm. **c** $^{230}Th_{ex}$ record with extinction age of 290 ka highlighted as green triangle. **d** Relative siderite content expressed as ratio of the XRD siderite peak intensity vs. sum of total analyzed intensities (see "Methods" for details). **e** Inorganic carbon (IC) content. IC maxima representing increased contents of calcite and dolomite are marked by blue and pink triangles, respectively (see Supplementary Fig. 11 for details). **f, g** Relative smectite and kaolinite contents expressed as ratio of the XRD smectite and kaolinite peak intensities, respectively,

vs. sum of total analyzed intensities (see "Methods" for details). **h** Ratio of the (clino-) pyroxene/amphibole XRD peak intensities. The XRD records are shown as three-point moving average values. The OC-1 to OC-7 intervals highlighted as gray bars, are generally characterized by elevated contents of siderite, smectite, kaolinite, and pyroxene, the absence of sand, and absolute minima (values < 0) in $^{230}Th_{ex}$. The pink arrow marks the prominent pink layer (PL) (see Fig. 2 and Supplementary Fig. 17). The Last Glacial Maximum (LGM), characterized by OC-poor sediments, is highlighted by a purple bar. Source data of (**b–h**) are provided as a Source Data file.

## Detrital sediment provenance, transport processes, and ice-sheet history

Remaining questions dealing with the input of detrital TerrOC-rich sediments are (i) from where this material did come from, (ii) what were the main transport pathways, and (iii) how these processes are related to ice-sheet dynamics. Due to the complex geology of the circum-Arctic Ocean hinterland specific mineral assemblages determined in Arctic marine sediment cores may allow for identifying source areas of the detrital sediments (Fig. 1)[51–56]. For example, elevated dolomite concentrations are indicative for a North Canadian source region as well as Svalbard, and high concentrations of amphiboles (in combination with high illite values) are indicative for Siberia and the adjacent East Siberian Sea (Fig. 1). Eurasian source areas are characterized by input of sediments with high

concentrations of clinopyroxene, siderite, smectite and kaolinite (Fig. 1). Having this in mind and looking at the records of mineralogical tracers determined in Core PS115/2-2, it is obvious that the dark gray sediments of the OC-1 to OC-7 intervals display significant maxima in (clino)pyroxene, smectite, kaolinite and especially siderite (Fig. 6d–h), but obviously no dolomite (Supplementary Fig. 11a). The maxima in relative siderite concentration mirror the maxima in inorganic carbon content (Fig. 6e), suggesting that siderite is the predominant carbonate mineral (with only a few short-term events where calcite and dolomite, respectively, are the predominant carbonate minerals; Supplementary Fig. 11a). Most probable source areas of these sediments are Paleozoic and Mesozoic sedimentary rocks (clay-, mud and sandstones, carbonates, especially siderite) characterized by partly elevated contents of ancient/petrogenic OC,

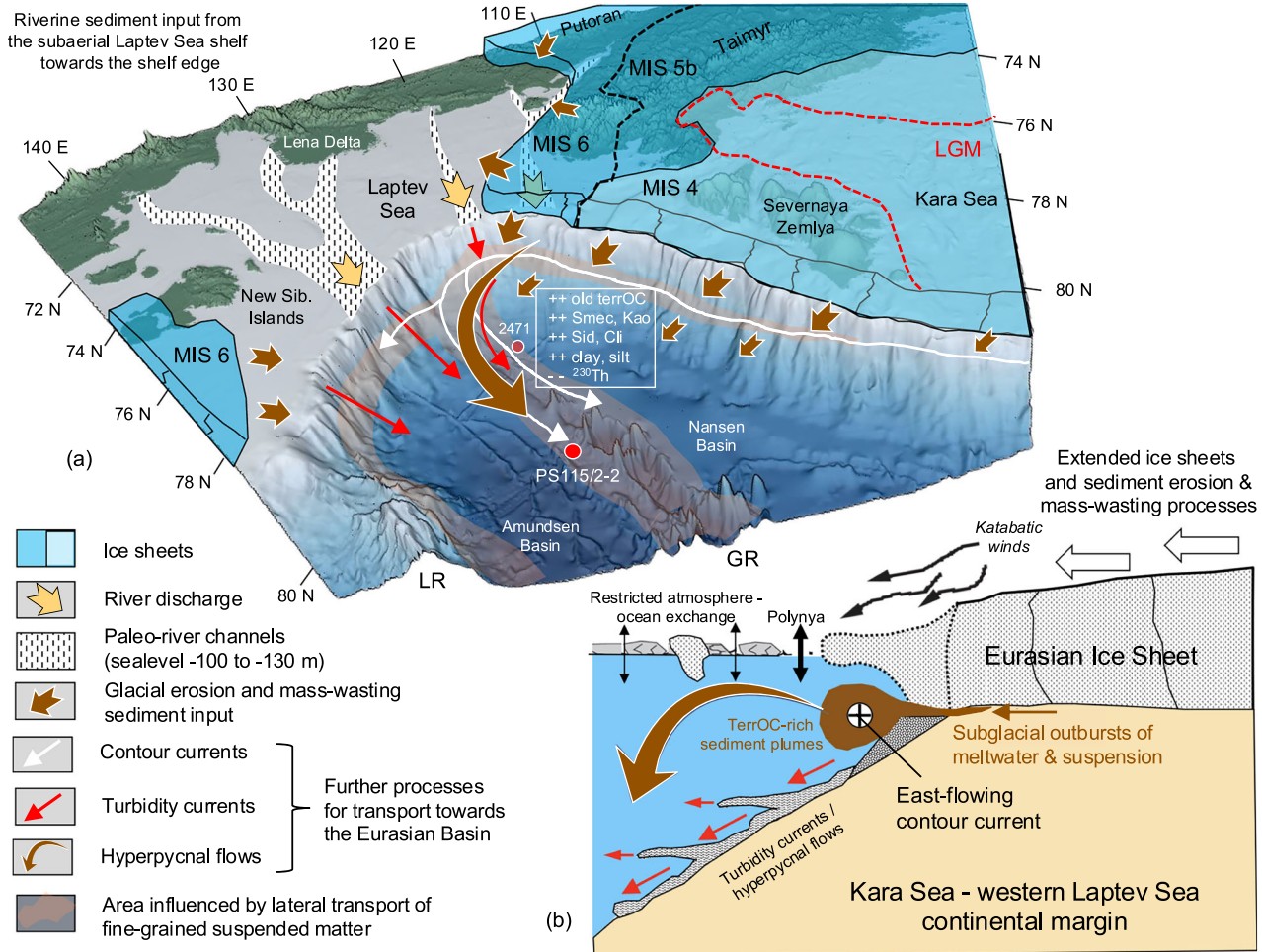

**Fig. 7 | Schematic illustration of the interrelationships between ice sheet dynamics, fresh-/meltwater discharge, and detrital sediment and OC input along the Kara-Laptev-Sea continental margin and adjacent Eurasian Basin.** **a** Three-dimensional cartoon figure showing the Eurasian ice sheet extent during MIS6, MIS 5b, MIS 4 and the Last Glacial Maximum (LGM) (for extent of ice sheets see ref. 8 and references therein), glacial erosion and mass-wasting sediment input across the shelf edge down-slope, and further long-distance transport of fine-grained suspension in turbidity currents, contour currents and hyperpycnal flows towards the Eurasian deep basins. Furthermore, the subaerial Laptev Sea with ancient river valleys provided further pathways for sediment transport across the shelf edge into the deep sea. Smec smectite, Kao kaolinite, Sid siderite, Cli (clino) pyroxene. The three-dimensional base map has been produced using the Fledermaus Software and the IBCAO Version 4.0 data[105]. **b** Cartoon figure of South-North transect across the Kara-Laptev-Sea continental margin with an extended Eurasian Ice Sheet during glacial intervals MIS 4, MIS 5b, MIS 6 (and MIS 8 and MIS 10 based on PS115/2 records) and related glacigenic processes such as subglacial outbursts of meltwater and suspension, and down-slope transport in turbidity currents and hyperpycnal flows. During these glacial intervals, the sea ice cover was also more extended with restricted atmosphere-ocean exchange, except for the area directly in front of the ice sheet where strong katabatic winds may have resulted in open-water polynya-type conditions. For further details and references, see text.

plant fragments and coals, and cropping out on Taimyr Peninsula and the West Siberian Basin (Supplementary Fig. 12)[57–61].

The OC-1 to OC-7 intervals correlating with glacials/cold intervals, were formed during times when the area around Taimyr Peninsula, Severnaya Zemlya, and the Kara Sea was covered by the extended EIS (Figs. 1 and 7a). With pulses of ice-sheet advance and retreat during glacial to deglacial times, ice-sheet erosion of sub-glacial OC-rich sedimentary rocks and soils in the Taimyr-Severnaya Zemlya-Kara Sea area delivered sediments that were "pushed" onto the shelf and further across the shelf edge towards the Eurasian Basin, accompanied by subglacial outburst of meltwater and suspensions[62]. For the EIS advance during MIS 4/3, these glacigenic processes are clearly reflected in diamictons, debris flows, and terminal moraines mapped in Parasound profiles off Severnaya Zemlya and Taimyr, as well as sediment Core PS2782 recovered in the Schokalsky Channel (Supplementary Fig. 13)[27,63]. Parts of the sediments may have also been delivered onto the subaerial Laptev Shelf and transported fluvially towards the shelf edge and beyond. Further down-slope sediment transport occurred via turbidity currents, sediments' rain-out from nepheloid layers and/or hyperpycnal flows, with coarse-grained material deposited more proximal whereas the fine-grained material may become available for a long-distance transport (Fig. 7)[64,65]. Such direct transport via turbidity currents is reflected in the sediments of the more coarse-grained turbidites in the more proximal Core PS2471 and the more fine-grained sediments in the distal Core PS115/2-2, both characterized by elevated TerrOC contents, high C/N ratios, and high kaolinite contents as well as minima in $^{230}Th_{ex}$ concentrations that can be correlated perfectly between both cores (Supplementary Fig. 14). The internal sedimentary structures of the glacial OC-rich intervals as well as grain-size proxies such as the Zr/Rb ratio also point to current-controlled sedimentation at Core PS115/2-2 (Supplementary Fig. 15).

## Discussion

In this study, we have developed a robust chronostratigraphic framework that allows for the first time to reconstruct in detail the interrelationship between ice-sheet dynamics and OC burial in the central

Eurasian Basin during the last 430 kyr, and to correlate marine and terrestrial records of ice-sheet history. Overall, we are confident that our results are of major interest for the broad Arctic scientific community and beyond, and that the well-dated Core PS115/2-2 might serve as a key core for future studies of Arctic paleo-climate reconstructions.

Applying organic-geochemical and mineralogical proxies we identified prominent well-defined intervals with strongly elevated concentration of ancient (petrogenic) OC and significant maxima in (clino)pyroxene, smectite, kaolinite and siderite, originated from the Taimyr-Severnaya Zemlya-Kara Sea region and correlating with glacial intervals MIS 10, MIS 8, MIS 6, MIS 5d/5b and MIS 4/3 and subsequent terminations (Figs. 4 and 6). During times of an extended EIS and lowered sea-level, huge amounts of OC-rich soils, sediments, and rocks have been eroded and transported by glacigenic processes onto the shelf and across the shelf edge, followed by down-slope turbidity currents and/or lateral current-induced transport of the fine-grained suspension. That means, the fine-grained OC-rich proportion of the sediments may have fed the eastward flowing contour currents along the Kara-Laptev-Sea continental margin transporting the suspension towards Site PS115/2-2 where the final OC burial occurred (Fig. 7).

The interpretation that the dark gray OC-rich intervals represent phases of EIS advance perfectly fits with the EIS reconstruction for MIS 6 to MIS 1 based on terrestrial field work (Fig. 8)[58]. Unfortunately, the traces of early shelf-based Kara Sea Ice Sheet glaciations in MIS 12–14 and MIS 8 are sparsely preserved in the terrestrial records[66], preventing a direct marine/terrestrial data comparison. In contrast to the older glacial intervals, no sediments with elevated contents of ancient/petrogenic OC were deposited at Site PS115/2-2 during MIS 2 (Fig. 4), when the EIS was more restricted and Severnaya Zemlya was not covered by ice (Figs. 7a and 8)[58,67]. This observation is further supporting our interpretation that glacial erosion followed by mass-wasting processes

probably were the main primary trigger for elevated OC burial rates at the core location (Fig. 7).

Pulses of suspension-loaded meltwaters during the EIS glacial advance and/or deglaciation have also been proposed from isotopic measurements on the detrital (Nd-Sr-Pb) and authigenic (Nd) phases in central Arctic Ocean deep-sea sediments off Lomonosov Ridge[65]. Similar processes have been described in multi-proxy sedimentary studies from the Ross Sea continental margin/Antarctica[68,69], the Antarctic Peninsula margin[70], and the Arctic Canada Basin[71]. That means, sediments delivered by turbidity currents or density plumes through fluvial and glacial processes, were deflected and deposited under the influence of contour currents[70,71]. Furthermore, such a scenario is supported by modern observations as well. Long-distance transport of OC-rich plumes along the Kara Sea–Severnaya Zemlya–Laptev Sea continental slope has been directly measured on vertical transects within the Nansen Basin using a CTD equipped with a fluoro- and turbidimeter and an underwater camera system[72]. On their pathway towards the east as dense contour currents, these plumes originated in the St. Anna Trough region and triggered by down-slope flowing Barents Sea Bottom Water, may expand and penetrate into deeper waters of the Eurasian Basin, and finally act as a significant OC sink[72].

At Site PS115/2-2, the distinctly increased glacial burial rates of predominantly ancient/petrogenic OC of about 15–30 mg cm$^{-2}$ kyr$^{-1}$ (in MIS 10 even >50 mg cm$^{-2}$ kyr$^{-1}$) are about three to six times higher than those determined for interglacial time intervals (<5 mg cm$^{-2}$ kyr$^{-1}$) (Fig. 4f). This finding is not just a local phenomenon. When looking at many sites in the Arctic Ocean it is obvious that glacial intervals are characterized by thick, dark gray OC-rich sediments related to glacial erosion on the surrounding continents and mass-wasting processes of sediment transport across the shelf edge towards the deep-sea basins (Supplementary Fig. 16; see also similar studies of sediment cores from the Ross Sea continental margin/Antarctic[68,69]), indicating that high-latitude deep-sea basins were

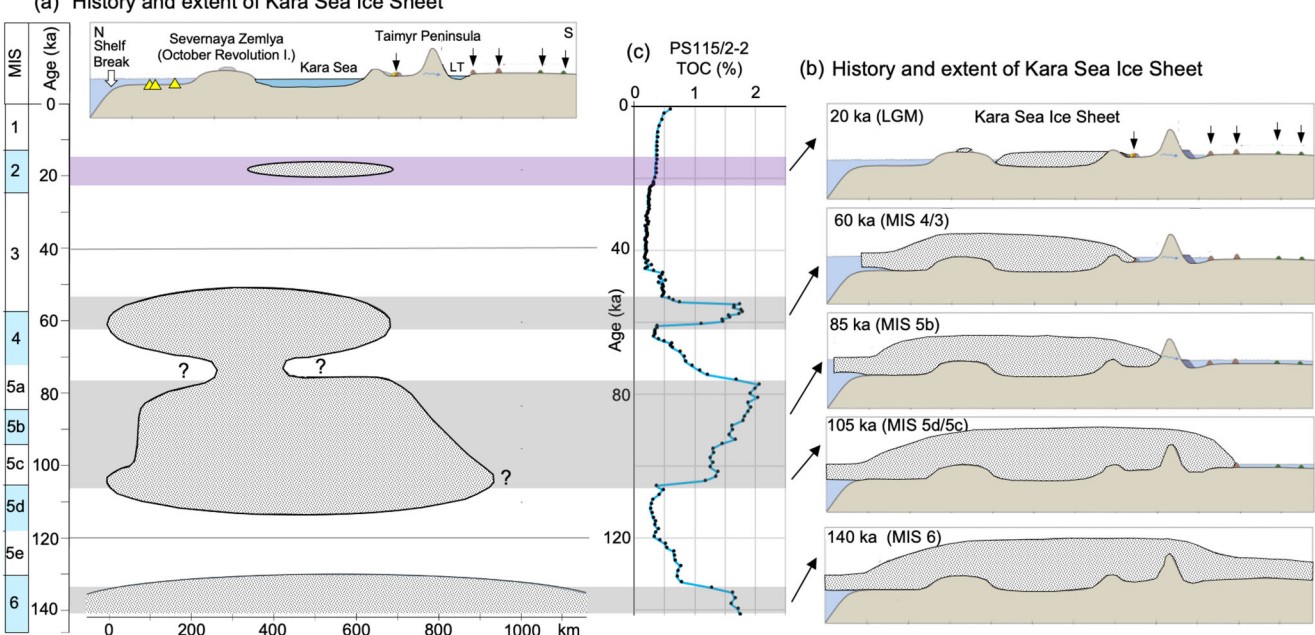

**Fig. 8 | Schematic illustration of the timing and extent of the Kara Sea Ice Sheet based on terrestrial data from Taimyr Peninsula and Severnaya Zemlya.**
**a** Glacial history with extended glaciations on Taimyr Peninsula, Severnaya Zemlya and the eastern Kara Sea during MIS 6, MIS 5d/5c, 5b and MIS 4/3 and a more restricted ice sheet during the Last Glacial Maximum (LGM), represented in **b** North-South profiles from the Kara Sea shelf break to the Arctic Ocean, over Severnaya Zemlya and the Taimyr Peninsula (from ref. 58, modified and supplemented). The southern extent of the ice sheet on Taimyr Peninsula is based on

mapping of terminal moraines shown as small black arrows in the North-South profile[58]. North of Severnaya Zemlya terminal moraines have been mapped during Polarstern Expedition ARK-IX/4, added as yellow triangles in the profiles (see Supplementary Fig. 13 for details and references. **c** TOC record for Core PS115/2–2 representing the last 140 kyr with the OC-3 to OC-1 intervals (this study). The timing of the OC-rich intervals in Core PS115/2–2 interpreted as glacial advances, perfectly coincide with intervals of maximum extent of the Kara Sea ice sheet based on terrestrial data, as highlighted by the gray bars.

probably important sinks for glacial OC burial. The dominant proportion of the OC fraction in these sediments is reworked and/or ancient/petrogenic TerrOC as shown in organic geochemical proxies[73,74]. Unfortunately, robust age models are often not available from these deep-sea sediment cores, preventing to calculate a more accurate basin-wide Arctic OC budget and estimating its significance within the global OC budget.

The elevated burial of TerrOC (Fig. 4f) may have supported the change in sedimentary redox conditions from oxic to anoxic. The downslope transport of huge amounts of EIS subglacial meltwater and (TerrOC bearing) sediments as high-density flows, i.e., from an environment that already might represent anoxic conditions[75], may further exaggerate suboxic/brackish conditions in the water column[62,65]. Anoxic sedimentary conditions are indicated by specific biomarkers, so-called $C_{33}$ dialkyl glycerol diether ($C_{33}$ DGD) lipids, that exclusively occur in the dark gray OC-rich intervals of kastenlot Core PS115/2-2-2 (Fig. 4e; Supplementary Fig. 9d). Such $C_{33}$ DGDs first identified in bacterial cultures[76] and to-date only known to be produced by anaerobic bacteria, predominantly sulfate reducers, were found in several different marine settings, and their abundance is reservedly correlated with oxygen content[77,78]. They are abundant in anoxic sediments and scarce or absent in oxic sediments and the water column, consistent with anaerobic bacteria being their only known source. Furthermore, these lipids are resistant to (oxic) diagenesis[79] and stable over geological time scales as they have been found in 32 Ma old marine sediments deposited under anoxic conditions[80]. Thus, the abundance of $C_{33}$ DGD in ancient sediments seems to be a perfect proxy for exploring past changes in sedimentary oxygenation[78]. Our interpretation of the $C_{33}$ DGD maxima in terms of sedimentary anoxic conditions is consistent with the simultaneous increase in uranium concentrations and the low pristane/phytane ratios of about 1 in the dark gray OC units (Supplementary Fig. 9c–f).

Due to the anoxic (sedimentary) environment, low but significant concentrations of more labile biomarkers indicative for open-water phytoplankton productivity and for sea-ice algae productivity are preserved in the glacial sediments as well (Fig. 4c, d and Supplementary Table 4). Such labile algae OC, however, is absent or only present in very minor concentrations in the beige-brown, highly oxidized interglacial sediments. These oxic sedimentary conditions during interglacials may be also reflected in the Fe/Al maxima (Fig. 4b). Because increased primary productivity in the glacial central Arctic Ocean characterized by more extended sea-ice conditions is less probable[7], it seems to be more realistic that the elevated algae-type (marine) OC (MarOC) concentrations are related to lateral transport originated from the marginal ice zones (e.g., polynyas; Fig. 7b)[7,64] to the core location and increased preservation under anoxic sedimentary conditions at the core location. That means, the contemporaneous increase of both the predominant terrestrial OC and the minor but still significant marine OC was laterally transported together towards the core location. Furthermore, the presence of the algae-type OC at Core PS115/2-2 is most important for the overall ice-sheet reconstruction. Its presence demonstrates that even during strong glacial intervals there must have been at least occasionally open-water (polynya-type) conditions along the Eurasian continental margin with marine and sea-ice algae productivity, indicating that not all parts of the Arctic Ocean were permanently covered by ice shelves throughout major glaciations during the last 450 kyr as already discussed for MIS 6 in earlier studies[6,7].

Whereas the prominent phases of elevated TerrOC input at Site PS115/2-2 were controlled by glacigenic and oceanographic processes primarily triggered by the EIS (Figs. 1 and 7), some coarser-grained dolomite-rich intervals may point to occasional sediment supply from the Canadian Arctic (Fig. 1, Supplementary Fig. 11a). The alternate origin that dolomite is related to a local source of Carboniferous carbonate rocks cropping-out on Northern Svalbard and laterally transported as fine-grained suspension in meltwater plumes and contour currents to our study site[51,55,81], can be excluded here for the coarse-grained dolomite-rich intervals. The most prominent pinkish horizon ("pink layer") with maximum dolomite content and extraordinary sand content of 35.6% occurs in the Core PS115/2-2 record at 556–562 cm depth and is dated to about 282 ka (i.e., late MIS 9/early MIS 8) (Fig. 6, Supplementary Figs. 11b and 17). Following widely published concepts, this suggests input of IRD from a North Canadian source[33,52,54,56,82]. That means this horizon might be correlatable with one of the very prominent wide-spread pink-white layers (most probably Pink-White Layer 2) described in many cores from the Amerasian Basin and even in some cores from the Eurasian Basin related to IRD input during times of an extended Laurentide ice sheet (LIS)[33,56,82–85] (see Supplementary Fig. 17), and thus be used as a key marker horizon for ocean-wide correlation and dating of Arctic sediment cores. Finding coarse-grained IRD with a Canadian provenance in Core PS115/2-2 indicates that the LIS has reached the shelf, and an extended Beaufort Gyre system allowed iceberg drifting across the entire Amerasian Basin towards the Eurasian Basin. Such an extreme situation may occur under anticyclonic conditions typical for negative phase of Arctic Oscillation (Fig. 1)[86], that also occurred on geological time scales as reflected in dolomite-enriched sedimentary records from the central Arctic Ocean (Supplementary Figs. 11 and 17; e.g., ref. 65 and references therein).

## Methods

### Basic shipboard and shore-based surveys and analyses

During Polarstern Expedition PS115/2, Core PS115/2-2-2 (81.2441°N, 123.4839°E; 3669 m water depth) and Core PS115/2-2-1 (81.2441°N, 123.4799°E; 3665 m water depth) representing near-surface sediments, were recovered at the same location by means of a Kastenlot (KAL) corer (764 cm recovery) and a box corer (46 cm recovery), respectively[21]. The KAL corer is a giant gravity corer with a rectangular cross section of 30 × 30 cm, and it has a penetration weight of 3.5 t and a core box segment sized 30 × 30 × 575 cm (Supplementary Fig. 2). The length of the KAL corer used was 11.5 m (=two connected boxes) plus 30 cm for the core catcher. The great advantage of the KAL is a wall-thickness of the barrel of only 2 mm. Because of the great cross-sectional area (900 cm²) and the small thickness of the barrels, the quality of the cores was generally excellent, and recovery equals penetration depth.

After opening the kastenlot boxes, the sediment surface was cleaned and photographs were taken, followed by sampling the sediment with large-sized plastic boxes of 1 m in length and 10 × 10 cm cross section, taking sediment slabs for x-radiographs, and ship-board sampling of discrete samples (Supplementary Fig. 2b). The sedimentary sections stored in these boxes were then used for visual core description, further photographs, multi-sensor core logging, and XRF scanning. A total of five sets of sub-cores are available from one kastenlot core, to be used for archiving and any kind of further detailed shore-based studies, including one complete set of sub-core sections stored deep-frozen for later biomarker studies (Supplementary Fig. 2b).

As the uppermost 11 cm of the Kastenlot Core PS115/2-2-2 was lost during the handling on deck (Supplementary Fig. 3), the top of the studied sedimentary sequence is represented by the near-surface sediment section of Core PS115/2-2-1. Geochemical data (e.g., $CaCO_3$ and TOC contents) and elemental ratios from XRF scanning (e.g., Mn/Al) produced shore-based in the AWI labs, allow a perfect correlation between both cores. Based on this correlation, the upper 34 cm of Core PS115/2-2-1 and the 34–760 cm section of Core PS115/2-2 were spliced together, resulting in the 760 cm thick complete and undisturbed composite section of Core PS115/2-2 (Supplementary Fig. 3).

For our study, a detailed continuous sampling of both cores PS115/2-2-1 and PS115/2-2-2 was carried out in the AWI sediment lab,

using 10 ml syringes (PS115/2-2-1: 22 samples; PS115/2-2-2: 376 samples) and 6.2 cm$^3$ plastic cubes (PS115/2-2-1: 19 samples; PS115/2-2-2: 323 samples) (Supplementary Fig. 2d). For the spliced composite sedimentary sequence of PS115/2-2, this results in totals of 381 and 328 samples, respectively. The syringes samples were freeze-dried, ground, and used for organic geochemical, radiogenic isotope, and XRD analyses, the plastic cubes were used for paleo- and rock magnetic measurements (see below for details). In addition, 144 large-sized samples of about 100 ml volume were taken every 5 cm, freeze-dried, weighed, and then sieved using a 63 μm sieve. The fraction >63 μm has been used for qualitative coarse-fraction analysis (e.g., determination of occurrences of foraminifers, coal fragments, detrital minerals, etc.).

## XRF scanning

The AWI Avaatech XRF (X-Ray Fluorescence) Core Scanner was used for rapid and non-destructive determination of the chemical composition of sediment sections of cores PS115/2-2-1 and PS115/2-2. Split core surfaces were scanned during a 10 kV, 30 kV, and 50 kV run, in order to obtain reliable intensities (counts per second) of specific major elements by analysing the spectra with for Arctic sediments adapted WinAxial™ models. In this paper, we only used Al, Mn, Fe, Rb, and Zr data, plotted and interpreted as elemental ratios Mn/Al, Fe/Al, and Zr/Rb. A more extended data set is available at https://doi.org/10.1594/PANGAEA.975853. For documentation and analytical details, we refer to the Avaatech Manual (https://epic.awi.de/id/eprint/37355/4/XRFCore-Scanner_user-manualV2.pdf).

## AMS$^{14}$C dating

AMS$^{14}$C datings were carried-out with the Accelerator Mass Spectrometry MICADAS system at the Alfred Wegener Institute in Bremerhaven. Three samples of the planktic foraminifer (*Neogloboquadrina pachyderma* sinistral) were taken from the box corer PS115/2-2-1 (Supplementary Table 1a). Using the radiocarbon calibration program CALIB 8.20 and the Marine 20 calibration curve with a global mean reservoir age ($R = 550$ years), and a local marine reservoir correction (deltaR) value of $345 \pm 60$ years, calendar ages were calculated. In addition, 17 samples of bulk AIOC residues were selected from upper 180 cm section for AMS$^{14}$C dating due to the lack of biogenic carbonate (Supplementary Table 1b). As the exact composition of the AIOC (i.e., marine vs. terrestrial OC, fresh vs. reworked OC) is not known, reservoir corrections have to be taken with caution. Thus, uncalibrated $^{14}$C ages of the AIOC are shown in Fig. 3 and calibrated ages (to be used cautiously) are only included in Supplementary Table 1. For further details including references see Supplementary Table 1.

## Magnetic properties and magnetostratigraphy

Paleo- and rock magnetic measurements were conducted at the paleomagnetic laboratory at the Faculty of Geosciences, University of Bremen. Paleomagnetic directions and intensities of NRM, anhysteretic remanent magnetization (ARM), generated in a peak alternating field of 100 mT and a biasing DC field of 50 μT, as well as isothermal remanent magnetization (IRM), imparted by applying 23 steps in DC fields from 0 to 700 mT using the internal pulse coil, were measured on an automated superconducting rock magnetometer (model 2G Enterprises 755HR)[87]. Low-temperature measurements were conducted at the University of Bremen using a Quantum Design MPMS XL-7. RPIs of the Earth's magnetic field were calculated using the so-called "slope-method" or pseudo Thellier method[88]. RPI was computed as the slope of the regression line of NRM intensities plotted versus the intensities of ARM and IRM for AF demagnetization levels 20 to 40 mT and 20 to 70 mT, respectively. Both RPI curves, RPI$_{arm}$ and RPI$_{irm}$, show almost similar variations (Supplementary Fig. 5g, h) and appear to be robust. The maximum angular deviation[89] varies between 0.3 and 12.5° with a mean of 3.5° (Supplementary Fig. 5b), which, in the context of Quaternary marine sediments from the Arctic Ocean, supports the

reliability of RPI data of Core PS115/2-2 for chronostratigraphic applications.

## Radioisotope analyses ($^{230}$Th, $^{231}$Pa, U isotopes)

The radioisotopes $^{230}$Th, $^{232}$Th, $^{231}$Pa, $^{234}$U, and $^{235}$U ($^{238}$U) were analysed in several batches over more than 2 years, with a slight change in separation methods during this period. All samples were digested in a microwave-assisted pressurized digestion using a CEM Mars Xpress® system equipped with an evaporation accessory. About 75 mg of sediment was digested together with isotopic spikes $^{229}$Th, $^{233}$Pa, and $^{236}$U in a mixture of 0.5 mL HF suprapur ®, 2 mL distilled concentrated HCl, and 3 mL distilled concentrated HNO$_3$. Each batch comprised the certified reference materials Irish Sea 385 (for radioisotopes)[90] and MESS-4 (for multielement analyses; provided by the National Research Council Canada), as well as procedural blanks.

Most samples were measured for Th and U isotopes using a previously published method[47] using only TRU resin in a prepFAST system (Elemental Scientific). Later in the procedure, the method was expanded to include Pa. Differing from the previous method, 50 μL of a 50 g/L iron solution was added to the full digestion, precipitated as Fe(OH)$_3$, reduced in volume and taken up in and taken up in 2 mL 10 M HCl after repeated evaporation steps. This solution was passed over a BioRad AG1 X8 2 mL column in the prepFAST system, eluting first Th with 9 M HCl, then Pa with 9 M HCL + 0.13 M HF, then Fe with 5 M HNO$_3$ (fraction discarded), then U with 0.01 M HCl (collected together with the Th fraction). Later, the Pa fraction was taken up again in 9 M HCl after repeated evaporation steps to eliminate HF. This solution was then passed another time over the prepFAST system equipped with a 2 mL BioRad AG1 X8 column to further separate Pa and Th, avoiding tailing and interferences from $^{232}$Th on the $^{231}$Pa and $^{233}$Pa isotopes. The combined Th+U fractions were passed over the prepFAST system with TRU resin like the previous batches, separating Th from U. All fractions were finally converted to 1 M HNO$_3$ and analyzed on an Element2 sector-field ICP-MS equipped with a Jet interface. U isotopes were analyzed using a cyclonic spray chamber, thorium isotopes and Pa isotopes using an Aridus II system for best sensitivities (typical sensitivity ca. 10–20 million cps/ppb U). U and Pa isotopes were analysed in low resolution ($R = 300$), Th isotopes in a special intermediate resolution of $R = 2000$, offering flat peaks for precise isotope ratios together with an improved abundance sensitivity compared to $R = 300$. $^{238}$U was calculated from $^{235}$U in order to avoid the intense ion beam of $^{238}$U on the detector. For the certified reference material IAEA-385, we found concentrations of $1810 + -0.037$ dpm/g $^{238}$U ($n = 31$), $1792 + -0.051$ dpm/g $^{230}$Th ($n = 32$) and $0.1079 + -0.0072$ dpm/g $^{231}$Pa ($n = 18$). The certified values of IAEA-385 are $1.74 + -0.06$ dpm/g for $^{238}$U and $1.89 + -0.108$ for $^{230}$Th (ref. 90). $^{231}$Pa is not certified in IAEA-385, but its parent $^{235}$U is specified at 0.081 dpm/g.

$^{230}$Th$_{ex}$ was calculated as the activity of $^{230}$Th in excess over the activity of its progenitor $^{234}$U, in line with previous publications on U-series data in Arctic sediments[31,41,91]. We note that several sections of the core show $^{230}$Th$_{ex}$ values below zero, and we also note signs of oxygen deficient conditions that led to the enrichment of authigenic uranium and possibly a secondary mobility of uranium in parts of the core as some oxygen is supplied from neighboring sections. $^{234}$U/$^{238}$U values far in excess of modern open ocean values (activity ratio 1.14) are seen in some deep parts of the sediment core. Our age constraints therefore rely on parts of the record where excess $^{230}$Th and $^{231}$Pa are seen in the absence of uranium enrichments or secondary mobility.

## Organic-geochemical bulk parameters

For the measurement of bulk parameters by means of elemental analysis and Rock-Eval pyrolysis, freeze-dried and homogenized sediments were used. Total organic carbon (TOC) contents were measured by Carbon-Sulfur Analyser (CS-125, Leco) after removing carbonate with hydrochloric acid. Total carbon (TC) and total nitrogen (TN)

contents were determined by Carbon-Nitrogen-Sulfur Analyser (Elementar III, Vario). Inorganic (carbonate) carbon (IC) was calculated as IC = TC − TOC. The C/N ratios used as a first-order proxy for estimating the marine (C/N = 5–8) and terrestrial (C/N >> 10) proportions of the OC (see ref. 12 and references therein), were calculated as "TOC/TN" ratio. As the TN values in sediments from the Laptev Sea continental margin and adjacent deep sea as well as the sediments from Core PS115/2-2 may also contain significant amount of inorganic nitrogen, the C/N ratios of Core PS115/2-2 are certainly minimum values, i.e., when corrected for inorganic nitrogen the maxima of 14–16 would increase to 25 and more (Supplementary Fig. 9)[10]. Based on dry bulk density values and linear sedimentation rates, bulk sediment accumulation rates and, by considering TOC values, bulk TOC accumulation rates were calculated. For calculation procedure, we refer to https://doi.org/10.1594/PANGAEA.975805.

Rock-Eval pyrolysis was performed on bulk ground sediment samples using a Rock-Eval 6 analyzer[92–95] to determine (1) the amount of hydrocarbons (HC) present and generated by pyrolytic degradation of the OC during heating up to 650 °C, (2) the amount of carbon dioxide ($CO_2$) generated from the decomposing OC during heating up to 390 °C, and (3) the temperature of maximum pyrolysis yield (Tmax). The measured temperature of maximum (peak) generation of HCs during pyrolysis represents "Tpeak", the corrected value after apparatus vs. standard calibration represents "Tmax" (Tmax = Tpeak − 36°; see Fig. 5). The HC and $CO_2$ yields were normalized to OC and expressed as hydrogen index (HI) in mgHC/gOC and oxygen index (OI) in mgCO$_2$/gOC, respectively. Plotted in a HI vs. OI ("van Krevelen-type") diagram, this allows a classification of the OC types. In immature sediments, HI values of <100 mgHC/gOC and high OI values of >>100 mgCO$_2$/gOC are generally typical of terrigenous and/or more refractive/oxidized organic matter (kerogen type III and IV), whereas HI values of 300–600 mgHC/gTOC are typical of marine OC (kerogen types I and II) (Fig. 5). Extremely high OI values for bulk sediments (>>300 mgCO$_2$/gOC), i.e., values also measured in samples from Core PS115/2-2 (Fig. 5a), might be an artifact due to the cracking at low temperature of carbonate minerals (especially siderite) that interferes with the determination of OC-derived $CO_2$ (cf., refs. 94,95). As proxy for the OC maturity, i.e., to identify ancient (petrogenic) OC, the Tmax values were used (Fig. 5b–e). Tmax values < 435 °C are indicative for immature OC, whereas mature (ancient/petrogenic) OC has Tmax values between about 435 and 475 °C. In combination with HI values, Tmax values may give further information about the composition as well as the maturity of the organic matter (cf., Fig. 5b).

## Biomarker analyses

For biomarker analysis, 5 g of freeze-dried and homogenized sediments were used for extraction by ultrasonication with dichloromethane/methanol (DCM/MeOH, 2:1 v/v). The internal standards 7-hexylnonadecane (7-HND, 0.076 µg), 9-octylheptadec-8-ene (9-OHD, 0.1 µg), 5α-androstan-3β-ol (androstanol, 10.8 µg), and squalane (3.2 µg) were added prior to the extraction. The extracts were concentrated and separated into hydrocarbon and sterol fraction by open silica gel column chromatography using 5 ml n-hexane and 9 ml ethylacetate/n-hexane, respectively. Furthermore, the sterol fraction was derivatized with 200 µl bis-trimethylsilyl-trifluoroacet-amid (BSTFA) (60 °C, 2 h). Highly branched isoprenoids (HBIs) were analysed by gas chromatography/mass spectrometry (GC/MS) (Agilent 7890GC-Agilent 5977 A), whereas n-alkanes, pristane, and phytane were analyses by GC (Agilent 6890GC). Sterols and C$_{33}$ dialkyl glycerol diether lipids (DGD, consisting of two C$_{15}$ alkyl chains) were analyses by GC/MS (Agilent 6850GC-Agilent 5975 A). Compound identification was based on comparison of GC retention times with reference compounds and published mass spectra for HBIs, sterols, and DGDs. IP$_{25}$, HBI II, and HBI III (Z) were quantified based on their molecular ions m/z 350, m/z 348, and m/z 346, respectively, in relation to the fragment ion

m/z 266 of the internal standard (7-HND). The ratio of the related response factors is nearly 1:1:1. Brassicasterol (24-methylcholesta-5,22E-dien-3β-ol), campesterol (24-methylcholest-5-en-3β-ol), sitosterol (24-ethylcholest-5-en-3β-ol), dinosterol (4a-23,24-trimethyl-5a-cholesta-22E-en-3β-ol), and DGDs were quantified as trimethylsilyl ethers. The molecular ions m/z 470, m/z 472, m/z 486, and m/z 500 for the sterols, and the fragment ion m/z 130 for DGDs were used in relation to the molecular ion m/z 348 of the internal standard androstanol. External calibration curves and specific response factors were applied for balancing the different responses of molecular ions of the analytes and the molecular/fragment ions of the internal standards. For quantification of n-alkanes, pristane, and phytane, squalane was used as internal standard. The sum of short-chain n-alkanes was calculated using C$_{17}$ and C$_{19}$ compounds, whereas the sum of long-chain n-alkanes was calculated using C$_{27}$, C$_{29}$, and C$_{31}$ n-alkanes. The carbon preference index (CPI) was calculated based on the concentrations of the long-chain n-alkanes C$_{24}$ to C$_{34}$, using the following equation:

$$CPI = 0.5 * \left( \left( \frac{C25 + C27 + C29 + C31 + C33}{C24 + C26 + C28 + C30 + C32} \right) + \left( \frac{C25 + C27 + C29 + C31 + C33}{C26 + C28 + C30 + C32 + C34} \right) \right)$$

For analytical details and further references see refs. 96,97 and Supplementary Fig. 9. All biomarker data are summarized in Supplementary Table 4.

## XRD analyses of bulk sediments

Dried bulk samples were ground to a fine powder (<20 µm particle size) and prepared with the Philips backloading system. A standard deviation of ±5% can be considered as a general guideline for mineral groups with >20% clay fraction. In addition, the determination of well crystallized minerals like quartz, calcite, or aragonite can be done with better standard deviations (±1–3 %)[98].

The samples were measured on a Bruker D8 Discover diffractometer equipped with a Cu-tube (k$_\alpha$ 1.541, 40 kV, 40 mA), a fixed divergence slit of ¼° 2θ, a 90 samples changer, a monochromatisation via energy discrimination on the highest resolution Linxeye detector system. The measurements have been done as a continuous scan from 3 to 65° 2θ, with a calculated step size of 0.016° 2θ. Mineral identification has been done by means of the Philips software X'Pert HighScore™ Version 1.2 (ref. 99), and identification of sheet silicates have been done with the freely available Apple MacIntosh X-ray diffraction interpretation software MacDiff 4.25 (ref. 100). This was followed by peak identification and determination of intensity ratios[101]. The relative contents of smectite and kaolinite as well as the carbonate minerals siderite, dolomite, calcite, and aragonite are expressed as ratio of the XRD single mineral peak intensity vs. sum of total analyzed intensities (see Fig. 6 and Supplementary Fig. 11). The entire list of minerals determined by XRD (including quartz, feldspars, heavy minerals etc.) that are used to calculate the sum of total analyzed intensities are available at https://doi.org/10.1594/PANGAEA.975865.

## Data availability

All data sets generated within this study and included in this published article (and its supplementary information files) are available at https://doi.org/10.1594/PANGAEA.975790. Source data are provided with this paper.

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

## Acknowledgements

For technical assistance, we gratefully thank Walter Luttmer (AWI Bremerhaven: biomarker and Rock-Eval analyses), Sharon Dbritto and Valea Schumacher (AWI Bremerhaven: C-N-S elemental analyses), Ingrid Stimac and Denise Bethke (AWI Bremerhaven: radiogenic isotope analyses), Johannes Birkenstock and Ella Schmidt (Crystallography and Materials Research Group, Faculty of Geosciences, Bremen University: XRD analyses), and Gerhard Bohrmann and Petra Witte (Faculty of Geosciences, Bremen University; SEM photographs & EDAX analyses). The study used samples and data provided by AWI (Grant No. AWI_PS115.2_01). Funding by the German Research Foundation (DFG) is gratefully acknowledged (grant STE 412/30).

## Author contributions

R.S. developed the concept of the study, was chief scientist of the PS115/2 Expedition, developed the lithostratigraphic framework, did the evaluation and interpretation of the Rock-Eval data, and wrote the first version of the manuscript. R.S., T.F., and W.G. developed the age model. T.F. conducted the rock- and paleomagnetic measurements and interpretations. K.F. conducted the biomarker analyses, evaluation, and quality control. W.G. led the elemental and radioactive isotope analyses and evaluation. J.M. carried out the XRF analyses and evaluation. F.N. compiled the physical property and Parasound data. C.V. conducted the XRD measurements and evaluation. C.S. produced a first low-resolution organic-carbon record during her master thesis. E.B. produced the basic 3D map used for the schematic illustration of Fig. 7. R.S., T.F., K.F., W.G., J.M., F.N., and C.V. contributed to the data interpretation and writing of the final version of the manuscript.

## Funding

## Competing interests

The authors declare no competing interests.
