## [Transparent Peer Review file · Nature Communications]

A 430 kyr record of ice-sheet dynamics and organic-carbon burial in the central Eurasian Arctic Ocean

Corresponding Author: Professor Ruediger Stein

Version 0:

Reviewer comments:

Reviewer #1

(Remarks to the Author)

This study of one deep sea sediment core from the Arctic Ocean off Siberia is a very nice study of sediment flux variability associated glacial/interglacial ice sheet forcing. The chronology is well developed and consistent among the different dating techniques. I think this is a very nice and well developed paper.

Reviewer #2

(Remarks to the Author)

Manuscript NCOMMS-24-59102

Title: A 430 kyr record of ice-sheet dynamics, sediment discharge, and organic-carbon burial in the central Eurasian Arctic Ocean

Authors: Ruediger Stein, Thomas Frederichs, Kirsten Fahl, Walter Geibert, Jens Matthiessen, Frank Niessen, Christoph Vogt, Cynthia Sassenroth, Evgenia Bazhenova

The manuscript reports about the interrelationship between ice-sheet dynamics and Organic Carbon (mainly of continental origin) burial in the central Eurasian Basin during the last 430 kyr.

The work is based on a very large (impressive), multi-proxy dataset collected on two complementary sediment records (1 Kastens core and 1 box core), and the data were compared with previous studies conducted on other cores in the area. The multi-proxy approach allowed for a very good and solid stratigraphic reconstruction of the new studied cores and correlation with previously studied cores.

The work outlines the huge delivery of organic matter during glacial stages and glacial terminations and present strong evidence of possible anoxic conditions during deposition.

I have two main concerns about the manuscript for which I ask a moderate revision:

1) The method used for the AMS radiocarbon data calibration.

There are not information about the software that was used for the radiocarbon calibration (Calib software?). The global mean reservoir should be already considered by the software, whereas an additional local reservoir effect should be considered by a DeltaR +/- error, that is not mentioned by the authors. Also, why the authors considered to use the 2-sigma values whereas the most used are the 1-sigma? Alternatively, the authors may want to consider the use of the Median Probability given by the (Calib) calibration software. In any case, it would be better to report all the calibration information on Table 1 of the Supplementary (1- and 2-sigma and the Median Probability)

2) The interpretation about the origin of the pink layers and, therefore, about the paleo ice sheet dynamic and the palaeoceanographic structure of the paleo ocean currents.

Have the authors considered the work of Rasmussen and Thomsen from 2013 (Pink marine sediments reveal rapid ice melt and Arctic meltwater drain during Dansgaard–Oeschger warmings, *Nature Comm.*) which discusses the presence of pink layers in the sedimentary sequence recovered from the north-western margin of Svalbard?

In Rasmussen and Thomsen (2013) the pink layers derive from erosion of the Devonian Red Beds composed mainly of conglomerates and sandstones that outcrop in the North-central area of Spitsbergen. The geology of Svalbard also indicates the presence of carbonates (including dolomite) of Carboniferous age in the same area (Piepjohn, K. and Dallmann, W.K., 2014, *Stratigraphy of the uppermost Old Red Sandstone of Svalbard*

(Mimerdalen Subgroup), *Polar Research* 2014, 33, 19998; Manby, G. and Lyberis, N., 1992, *Tectonic evolution of the Devonian Basin of northern Svalbard*. *Norsk Geologisk Tidsskrift*, Vol. 72, pp. 7-19, JSSN 0029-196X).

Perhaps the authors may consider and discuss an alternative origin of the pink layers recovered in the studied record, which considers a more local origin: deposition of fine-grained sediments suspended in meltwater plumes released into Woodfjorden and Widgefjorden (north of Spitsbergen) during the retreat of the paleo-ice streams that eroded the Devonian Red Stones and Carboniferous carbonates (including dolomite) outcropping in the nearby inland.

METHODS

Radiocarbon calibration (Table 1 of supplementary), which software was used for the calibration? Calib? The global mean reservoir should be already considered by the software, whereas an additional local reservoir effect should be considered by a ΔR +/- error.

Why the authors considered the 2-sigma values when the most common is to consider the 1-sigma? Alternatively, the authors may want to consider the Median Probability given by the calibration software (Calib)

XRD analysis, the method describes the use of the entire sample (bulk XRD analysis), however the percentages of the clay mineral kaolinite is used for the discussion of the provenance of the sediment. Did you extract the kaolinite content from the bulk sediment analysis, or did you also perform analysis on the clay fraction alone? If the latter, please describe the sample preparation, and report also the information about the other clay minerals.

SUPPLEMENTARY

Tables 1 to 4 looks like a snapshot of an xls spreadsheet and they are not easy to read, please make a proper table.

Table 2 Suggested some changes in the caption

FIGURES

Figure 1, The table reported in figure 1 is too small

Figure 2, The core location in figure 2a is difficult to see, please change colour of the marker

Figure 3, Nice hand-made core logs, however the details are difficult to see (also possibly too many). Please digitalize the core logs

Figure 10, Legend difficult to read

Figure 14, A core log legend and a simple map are missing

Additional comments are reported in the attached document.

Best regards, Renata G Lucchi

Reviewer #3

(Remarks to the Author)

The authors found that terrOC deposition increased during glacial (stadial) periods in the deep Eurasian Basin. These terrOC was transported by ice sheet advance ("pushed") across the shelf edge, then by mass-wasting processes to the down slope area. In addition, these OC is glacially eroded from subglacial sedimentary rocks and soils in the Taimyr-Severnaya Zemlya-Kara Sea area. Glacial OC burial rates are 3-6 times higher than interglacial times and the anoxic condition improved preservation of more labile algae-type OC. Therefore, they argued that Arctic deep-sea sediments were an important OC (and CO₂) sink during glacial periods. The proposal is interesting and the records show clear patterns; increased values in the dark layers and decreased values in the brown layers for all records. However, there are some issues should be solved.

1. chronology

Chronology is one of hot issues in the Arctic Ocean; lithostratigraphic chronology vs magnetic property based chronology (e.g., RPI and/or magnetic reversals). 5 pages of 16 pages of main text are about justification and how the age model was set. As a result, more attention is likely to be drawn to the chronological part rather than the key findings of this study. Why the proportion of chronological part is significant in this manuscript is related to the current age constraining issue. In this sense, I cautiously suggest that this manuscript might be more suitable for more specific journals rather than *Nature Comm.* which deals with broad interests rather than local/regional ones.

2. glacial erosion of subglacial sedimentary rocks and soils in the Taimyr-Severnaya Zemlya-Kara Sea areas?

High C/N ratios and high Tmax support that OC is reworked and have terrestrial characteristics. Organic-geochemical and mineralogical proxies showed significant maxima in (Clino)pyroxene, smectite, kaolinite and siderite in high OC dark intervals originated from the above region. Subglacial sedimentary rocks and soils in that area are the source for high OC-rich dark layer. Sedimentary rocks mean (semi)consolidated form? After glacial erosion, chips or fragments of mudstone are

expected to be found rather than mud form. However, very fine grain size are characteristics for the dark layers. Ancient (patriotic) OC has lower T_{max} than relatively recently deposited brown layer sediments' OC? Is there any evidence for ancient OC? Such as microfossils, or some specific minerals found in the far past? In the Ross Sea, continental slope sediment study showed increased biogenic opal, TOC, and C/N (Ha et al., 2022) and a deep-basin sediment study also showed increased diatoms (reworked) in laminated mud during glacial periods (Bollen et al., 2022). They also interpreted that they are transported by turbidity current as ice sheet advanced across the continental shelf edge. But, Ha et al. (2022) suggested that the sediments deposited in the slope during glacial periods were originally deposited on the shelf during former interglacials.

According to the authors, the dark layer sediments are a result of contour and turbidity currents (upward fining supports this). This means, they are laterally transported and likely sedimentation rates are high. However, marine and sea ice algae immediately increased as if they were already in the sediment. The strong correlation between TOC and all other proxies suggests that they were transported together rather than due to better preservation under anoxic conditions.

If authors want to argue that increases in marine and sea ice algae are due to surface biological activity under glacial polynas conditions, how much OC is derived from these glacial surface biological productions compared to terrOC should be calculated. This is important because authors also argued that the Arctic Ocean played a role as a sink for CO₂.

3. OC-3,-5,and -6 covers full glaciation to deglaciation phases. However, there is no difference between glacial part and deglacial part in the records although more subglacial melt water is likely expected during deglacial (retreating phase). How about outburst of meltwater and suspension changes according to ice sheet phases should be considered.

4. Although authors argued 3-6 times more OC burial occurred in the Arctic Ocean during glacial periods, 0.015-0.03g/cm²/kyr is not outstandingly high, but is relatively low compared to other regions. Thus, the increased OC burial during glacial periods are difficult to be thought as a major significance in the Earth's climate system.

More importantly during glacial periods continental margin deep sea areas receives huge amount of reworked sediment input from continental shelf regions as a result of sea level drop. In the Antarctic Ocean, Ross Sea and Amundsen Sea (+West Antarctic Peninsula) also showed ice sheet dynamics associated OC burial during glacial periods (high OC, C/N in dark laminated mud with higher sedimentation rates than this study). Thus, the clear peaks of OC accompanying with distinct other proxies in the Arctic Ocean during glacial periods may not be a significant phenomenon in a global point of view.

In summary, the clear cyclicity of lithology and OC, C/N, and all other biomarkers is interesting. However, why OC burial in the Arctic Ocean is important is not well convinced. In addition, its impact seems to be far more exaggerated, particularly global sink for OC and CO₂.

Bollen et al. 2022. Pleistocene oceanographic variability in the Ross Sea: A multiproxy approach to age model development and paleoenvironmental analyses. *Global and Planetary Changes* 216.

Ha et al. 2022. Glaciomarine sediment deposition on the continental slope and rise of the central Ross Sea since the Last Glacial Maximum. *Marine Geology* 445.

Hillenbrand et al., 2021. New insights from multi-proxy data from the West Antarctic continental rise: implications for dating and interpreting Late Quaternary palaeoenvironmental records. 257.

Version 1:

Reviewer comments:

Reviewer #2

(Remarks to the Author)

Manuscript NCOMMS-24-59102

Title: A 430 kyr record of ice-sheet dynamics, sediment discharge, and organic-carbon burial in the central Eurasian Arctic Ocean

Authors: Ruediger Stein, Thomas Frederichs, Kirsten Fahl, Walter Geibert, Jens Matthiessen, Frank Niessen, Christoph Vogt, Cynthia Sassenroth, Evgenia Bazhenova

The manuscript reports about the interrelationship between ice-sheet dynamics and Organic Carbon (mainly of continental origin) burial in the central Eurasian Basin during the last 430 kyr.

I have already reviewed this manuscript at the first round, and I can confirm this is a very interesting (thought impressive) work that identified key sites for stratigraphic, paleoceanographic and paleoclimatic reconstructions in the central Arctic but also dealt with OC burial at high-latitudes in deep-sea sediments during glacial stages / glacial terminations having a strong impact on the global OC and CO₂ budget in Earth's climate system. If I may, I would like to suggest a couple of articles in support of their discussion: Lucchi et al., 2013 that reports about high concentration of OC buried in glaciogenic diamicton on the western margin of Svalbard (Arctic) and Lucchi et al., 2007 that defined the concept of "glacial contourites" for barren, not bioturbated contouritic sedimentation deposited during glacial stages on the Pacific Margin of the Antarctic Peninsula (West Antarctica) as consequence of suppressed biologic activity induced by oxygen reduced/anoxic conditions during glacial periods.

The authors made a meticulous revision of the manuscript taking in consideration all the suggestion and concerns highlighted by the former reviewers. I made some additional, very minor, suggestions to the reviewed manuscript that are reported in the attached pdf file.

I consider the manuscript ready for publishing

Best regards
Renata G. Lucchi

Reviewer #4

(Remarks to the Author)

My role in this review is to comment upon authors responses/actions to previous reviewer #3 comments on the original manuscript, after this manuscripts´ revision. This is done on attached file.

Bremen, 20.12.2024

Prof. Dr. Ruediger Stein
Faculty of Geosciences
University of Bremen
Klagenfurter Str. 2-4
D-28359 Bremen, Germany

Subj.: Nature Communications manuscript NCOMMS-24-59102A

" A 430 kyr record of ice-sheet dynamics, sediment discharge, and organic-carbon burial in the central Eurasian Arctic Ocean "

Point-by-point response to the referees' comments

Following the numerous suggestions/comments/corrections given by the reviewers as well as the editor, we have carried-out a substantial revision of the manuscript. Furthermore, the Supplementary Information has been re-structured and divided into more logical "subchapters" (Supplementary Information No.1 to No.7). All our changes/corrections done in the manuscript text are highlighted by yellow background color and explained in the point-by-point response below.

(1) Comments of Reviewer #1 (in italics) and our/Authors' response

Reviewer 1: "This study of one deep sea sediment core from the Arctic Ocean off Siberia is a very nice study of sediment flux variability associated glacial/interglacial ice sheet forcing. The chronology is well developed and consistent among the different dating techniques. I think this is a very nice and well-developed paper."

Authors: Thanks for this very positive statement.

(2) Comments of Reviewer #2 (in italics) and our/Authors' response

Reviewer 2: "The work is based on a very large (impressive), multi-proxy dataset, and the data were compared with previous studies conducted on other cores in the area. The multi-proxy approach allowed for a very good and solid stratigraphic reconstruction of the new studied cores and correlation with previously studied cores. The work outlines the huge delivery of organic matter during glacial stages and glacial terminations and present strong evidence of possible anoxic conditions during deposition. I have two main concerns about the manuscript for which I ask a moderate revision"

Authors: Thank you very much for this very positive general statement highlighting the multi-proxy key approach with its main results – despite two main concerns we will comment on in the following.

Reviewer 2: Concern (1) "The method used for the AMS radiocarbon data calibration. no information about the software that was used for the radiocarbon calibration (Calib software?). ... local reservoir effect should be considered by a ΔR +/- error, that is not mentioned by the authors. why the authors considered to use the 2-sigma values whereas the most used are the 1-sigma? Alternatively,

the authors may want to consider the use of the Median Probability given by the (Calib) calibration software. In any case, it would be better to report all the calibration information on Table 1 of the Supplementary (1- and 2-sigma and the Median Probability)”

Authors: We are very grateful for this important comment, and we have to apologize that we did not give this information in the first version of the manuscript. Following Reviewer 2, all the missing information about the method used for the AMS radiocarbon data calibration, has been outlined under “Methods” (line 548-556) and, especially, in the legend of **Supplementary Table 1** (i.e., Calib 8.2 software, Marine 20 calibration, local reservoir correction, 1- and 2-sigma, Median Probability etc. as well as related references).

Reviewer 2: *Concern (2) “The interpretation about the origin of the pink layers. Rasmussen and Thomsen (2013) discuss the presence of pink layers in the sedimentary sequence recovered from the north-western margin of Svalbard.....Geology of Svalbard also indicates the presence of carbonates (including dolomite) of Carboniferous age in the same area (Piepjohn and Dallmann 2014; Manby and Lyberis 1992) authors may consider and discuss an alternative origin of the pink layers which considers a more local origin: deposition of fine-grained sediments suspended in meltwater plumes during the retreat of the paleo-ice streams that eroded the Devonian Red Stones and Carboniferous carbonates (including dolomite) outcropping in the nearby inland.”*

Authors: Thank you for this comment related to the origin of the pink layers. Reviewer 2 is right that (Carboniferous) carbonate rocks (including dolomites) are also cropping out on Northern Svalbard (we refer to Vogt et al. 2001 and have added Manby & Lyberis 1992 as suggested by Reviewer 2). We also agree that glacial erosion on Svalbard and input of reworked carbonates (dolomites) across the Svalbard shelf edge, followed by lateral transport of fine fraction within the intermediate (Atlantic) water mass along the northern Svalbard continental margin, is a possible process for sediment transport from Svalbard towards the east (as described and discussed in Vogt et al., 2001 and also Rasmussen & Thomsen 2013). However, this interpretation is not realistic for our situation. Here, we are referring to a single but very prominent “pink-layer event” characterized by very coarse-grained dolomite-rich sediments, i.e., a sediment fraction for which a long-distance transport in suspension can be excluded. Instead, discharge as ice-rafted debris via icebergs with a North Canadian source is the only realistic explanation and also supported by literature data (for details and references see revised manuscript line 466-486).

Reviewer 2: *XRD analysis, the method describes the use of the entire sample (bulk XRD analysis), however, the percentages of the clay mineral kaolinite is used for the discussion of the provenance of the sediment. Did you extract the kaolinite content from the bulk sediment analysis, or did you also perform analysis on the clay fraction alone?*

Authors: Thank you for this comment. We have to apologize that we did not outline our XRD approach and XRD records shown in relevant figures clearly enough. In our study, XRD analysis was carried out using bulk ground sediments. Thus, relative mineral contents are expressed as ratio of the XRD single mineral peak intensity vs. sum of total analyzed intensities (see Fig. 6 and **Supplementary Figs. 14b and 15**). In **Supplementary Fig. 14**, relative kaolinite contents determined in bulk sediments from Core PS115/2-2 (**Supplementary Fig. 14b**) are compared with published kaolinite contents determined in clay fraction samples from near-by Core PS2471-4 (**Supplementary Fig. 14a**). Thus, the latter kaolinite contents are expressed as percentage values (see “Methods” line 704-709 and **Supplementary Fig. 14** for details and references).

Reviewer 2: *Supplementary tables 1 to 4 looks like a snapshot of an xls spreadsheet and they are not easy to read, please make a proper table.*

Authors: Supplementary figures are shown in a more proper way. Furthermore, the tables will also be submitted separately as excel tables/spreadsheets.

Reviewer 2: *Comments to Supplementary figures*

Figure 1, The table reported in figure 1 is too small

Authors: Figure has been deleted.

Figure 2, The core location in figure 2a is difficult to see, please change color of the marker

Authors: Size and color of core locations have been changed (new Suppl. Fig. 1a).

Figure 3, Nice hand-made core logs, however the details are difficult to see (also possibly too many). Please digitalize the core logs

Authors: Core logs have been digitalized.

Figure 10, Legend difficult to read

Authors: Legend in the figure has been changed to make more readable (new Suppl. Fig. 12).

Figure 14, A core log legend and a simple map are missing

Authors: Core log legend has been added (new Suppl. Fig. 16); for core locations we refer to Fig. 1.

Reviewer 2 *marked additional smaller comments in the pdf attached to the review that we have considered/corrected in the revised version of the manuscript:*

Lines 303-306: Please rephrase between lines 305-307 to make more clear that the dark-gray intervals are the second type of OC-rich intervals indicated in line 302

Authors: Done - Line 296-300

Line 302: coal to continental-derived coal

Authors: Done - Line 306

Line 355: correct “end morains”

Authors: Done (changed to “terminal moraines” - Line 347)

Line 410: ... measured on vertical transects CTD? Give details

Authors: Done – “...using a CTD equipped with a fluoro- and turbidimeter and an underwater camera system...” (Line 405-406)

Line 412:triggered by Barents Sea Bottom Water, may expand and penetrate into deeper waters as well as into the sediment changed to “ – please correct.

Authors: Done - Line 408-409

(3) Comments of Reviewer #3 (in italics) and our/Authors' response

Reviewer 3: “The authors found that terrOC deposition increased during glacial (stadial) periods in the deep Eurasian Basin. These terrOC was transported by ice sheet advance (“pushed”) across the shelf edge, then by mass-wasting processes to the down slope area. In addition, these OC is glacially eroded from subglacial sedimentary rocks and soils in the Taimyr-Severnaya Zemlya-Kara Sea area. Glacial OC burial rates are 3-6 times higher than interglacial times and the anoxic condition improved preservation of more labile algae-type OC. Therefore, they argued that Arctic deep-sea sediments were an important OC (and CO₂) sink during glacial periods. The proposal is interesting and the records show clear patterns; increased values in the dark layers and decreased values in the brown layers for all records. However, there are some issues should be solved.”

Authors: Thank you very much for this positive general statement – despite several concerns we will comment on in the following.

Reviewer 3: Chronology is one of hot issues in the Arctic Ocean; lithostratigraphic chronology vs. magnetic property-based chronology (e.g., RPI and/or magnetic reversals). 5 pages of 16 pages of main text are about justification and how the age model was set. As a result, more attention is likely to be drawn to the chronological part rather than the key findings of this study. Why the proportion of chronological part is significant in this manuscript is related to the current age constraining issue. In this sense, I cautiously suggest that this manuscript might be more suitable for more specific journals rather than *Nature Comm.* which deals with broad interests rather than local/regional ones.

Authors: We totally agree with Reviewer 3 that “Chronology is one of hot issues in the Arctic Ocean” or, as described in our introduction (line 48-51) “A key problem in this debate about the history and significance of the Arctic cryosphere within the climate system, that has not been fully resolved yet, is the accurate dating of the Quaternary sediment sequences, a major challenge for the entire Arctic research community for several decades.” Core PS115/2-2 represents an undisturbed complete sedimentary sequence representing the last about 500 kyr, i.e., the last four glacial interglacial cycles (based on our age model). Such a well-dated 500 ka record was not available from this part of the central Arctic Ocean before our study. It allows for the first time detailed reconstructions of the history (timing) and dynamics of the Eurasian ice sheet history as well as the related OC burial history, i.e., a topic that is of overall interest also for a broad scientific community dealing with climate reconstructions in general. However, for this type of detailed climate reconstructions that would allow correlations with other Arctic, mid-/low-latitude and/or global climate records, a robust and data-supported age model is a fundamental prerequisite. Thus, a major part of our manuscript is devoted to this important topic. We realized that this subchapter became quite long and - based on the reviewer’s comment - we shorten the text and shifted parts as well as one figure into the supplement. However, we do not agree with the Reviewer’s opinion that because of focusing that much on the age model our study is not of broad interest (i.e., not suitable for a *Nature Communications*-type journal) but only is of local/regional interest. In contrast, we are quite confident that - because of the well-established and convincing age model - Core PS115/2-2 might become a key core also for future studies on reconstructing paleo-climate conditions that is of major interest for the entire Arctic scientific community and beyond.

Reviewer 3: Glacial erosion of subglacial sedimentary rocks and soils in the Taimyr-Severnaya Zemlya-Kara Sea areas. Subglacial sedimentary rocks and soils in that area are the source for high OC-rich dark layer. Sedimentary rocks mean (semi)consolidated form? After glacial erosion, chips or fragments of

mudstone are expected to be found rather than mud form. However, very fine grain size are characteristics for the dark layers.

Authors: As outlined in the text (line 376-383), we proposed that “During times of an extended EIS and lowered sea-level, huge amounts of OC-rich soils, sediments and rocks have been eroded and transported by glacial processes onto the shelf and across the shelf edge, followed by down-slope by turbidity currents and/or lateral current-induced transport of the fine-grained suspension. That means, the fine-grained OC-rich proportion of the sediments may have fed the eastward flowing contour currents along the Kara-Laptev-Sea continental margin transporting the suspension towards Site PS115/2-2 where the final OC burial occurred”. Processes of glacial erosion certainly resulted in very different grain-size fractions from very coarse-grained to very fine-grained particles (including chips and fragments of mudstone but also mud-/clay-sized material as typical for diamicton deposits). However, the coarser-grained particles accumulated more proximal whereas only the fine-grained mud was transported over long distances (e.g., in contour currents) to our core location (see also text line 353-362 and Supplementary Fig. 14).

Reviewer 3: *Ancient (patriotic) OC has lower Tmax than relatively recently deposited brown layer sediments' OC?*

Authors: Thanks for this comment related to the Rock Eval data, especially the Tmax values and the interpretation. The Tmax values shown in the original Figure 6 were misleading, and we have replotted the Rock Eval data in the new/revised Figure 5. For the beige-brown OC-poor sediments, extremely low hydrocarbon yields (S2 peak) were measured (Fig. 5c) and no Tpeak value was reached before the end of the pyrolysis at 650°C and thus no Tmax could be calculated (Fig. 5d and 5e). On the other hand, for the dark gray OC-rich intervals, high hydrocarbon yields (clear and prominent S2 peaks) were measured (Fig. 5c and 5e) and Tmax values of 435-440°C determined (indicating mature OC). Thus, the OC deposited in the brownish intervals are interpreted as highly degraded and/or oxidized (see Fig. 5).

Reviewer 3: *Is there any evidence for ancient OC? Such as microfossils, or some specific minerals found in the far past? In the Ross Sea, continental slope sediment study showed increased biogenic opal, TOC, and C/N (Ha et al., 2022) and a deep-basin sediment study also showed increased diatoms (reworked) in laminated mud during glacial periods (Bollen et al., 2022). They also interpreted that they are transported by turbidity current as ice sheet advanced across the continental shelf edge. But, Ha et al. (2022) suggested that the sediments deposited in the slope during glacial periods were originally deposited on the shelf during former interglacials.*

Authors: Unfortunately, we could not find any microfossils to be used for age determination. Our argument for ancient OC is coming (1) from the Rock Eval data (Tmax 435-440°C) and the occurrence of coal fragments and (2) the contents of specific minerals (siderite, kaolinite, smectite, pyroxene etc.) – all data supporting that most probable source areas of these sediments are Paleozoic and Mesozoic to Cenozoic sedimentary rocks (clay-, mud and sandstones and carbonates/siderite) characterized by partly elevated contents of ancient/petrogenic OC, plant fragments and coals, and cropping out on Taimyr Peninsula and the West Siberian Basin (see line 335-345). Processes of ice-sheet dynamics, glacial erosion, and sediment transport/deposition described for the Ross Sea continental margin (Bollen et al. 2022; Ha et al., 2022) have been considered/discussed in the revised manuscript (Line 399-403).

Reviewer 3: *According to the authors, the dark layer sediments are a result of contour and turbidity currents (upward fining supports this). This means, they are laterally transported and likely sedimentation rates are high. However, marine and sea ice algae immediately increased as if they were already in the sediment. The strong correlation between TOC and all other proxies suggests that they were transported together rather than due to better preservation under anoxic conditions.*

Authors: Thanks for this comment. Yes, we also agree that the terrestrial as well as the marine/sea ice algae OC were mainly transported by the same process (e.g., long-distance transport by turbidity and/or contour currents), as all the different OC proxies are strongly correlated. However, the original source of the terrestrial and algae marine/sea ice algae OC must be different (i.e., ancient terrestrial OC and - due to the biomarker proxies – more recent/Quaternary marine/sea ice algae OC). The presence of marine/sea ice algae OC point to open-water and ice-edge conditions. As even today, i.e., during interglacial conditions, primary productivity is very low and almost no marine OC is preserved in the interglacial sediments of the central Arctic Ocean, the chance to produce sediment with significant amounts of labile marine/sea-ice algae OC, is even worse during glacial times. That means, because increased primary productivity in the glacial central Arctic Ocean characterized by more extended sea-ice conditions is less probable, it seems to be more realistic that the elevated algae-type (marine) OC concentrations are related to lateral transport originated from the marginal ice zones (e.g., polynyas) that are known to occur in front of the ice sheet during glacial times. However, under normal oxic conditions, the labile marine/sea-ice algae OC cannot be preserved in the sedimentary record. Preservation of this type of labile OC is only possible under anoxic sedimentary conditions at the core location, probably caused by sediment-laden hyperpynal meltwater discharge and increased TerrOC burial (see manuscript line 426-430). This interpretation is exactly what our biomarker and ^{234}U records tell us (Fig. 4c-4e and Supplementary Fig. 9c-9f; manuscript line 446-458).

Furthermore, the presence of the algae-type OC at Core PS115/2-2 is most important for the overall ice-sheet reconstruction, a finding we have highlighted in the revised manuscript as one of the key results of our study. Its presence clearly demonstrates that even during strong glacial intervals there must have been at least occasionally open-water (polynya-type) conditions along the Eurasian continental margin with marine and sea-ice algae productivity in front of the ice sheet, indicating that not all parts of the Arctic Ocean were permanently covered by ice shelves throughout major glaciations during the last 450 kyr (line 458-464).

Reviewer 3: *If authors want to argue that increases in marine and sea ice algae are due to surface biological activity under glacial polynas conditions, how much OC is derived from these glacial surface biological productions compared to terrOC should be calculated. This is important because authors also argued that the Arctic Ocean played a role as a sink for CO₂.*

Authors: To give more exact estimates of the terrestrial and marine/sea ice algae OC is difficult (e.g., Stein and Macdonald 2004). Based on the Rock Eval data and kerogen microscopy data determined in OC of Cenozoic Arctic Ocean sediments, only rough first-order estimates are possible (e.g., Boucsein et al., 2002; Stein et al., 2006). From these data, only <10% of the OC is derived from marine/sea ice algae (Fig. 5 and legend line 1098-1101).

Reviewer 3: *OC-3,-5, and -6 covers full glaciation to deglaciation phases. However, there is no difference between glacial part and deglacial part in the records although more subglacial melt water is likely expected during deglacial (retreating phase). How about outburst of meltwater and suspension changes according to ice sheet phases should be considered.*

Authors: We are in line with this comment. That means, pulses of suspension-loaded meltwater and sediment may have occurred during the EIS glacial advance and/or deglaciation. Sediments delivered by turbidity currents or density plumes through fluvial and glacial processes, might have been deflected and deposited under the influence of contour currents (line 399-402).

Reviewer 3: *Although authors argued 3-6 times more OC burial occurred in the Arctic Ocean during*

glacial periods, 0.015-0.03g/cm²/kyr is not outstandingly high, but is relatively low compared to other regions. Thus, the increased OC burial during glacial periods are difficult to be thought as a major significance in the Earth's climate system. More importantly during glacial periods continental margin deep sea areas receives huge amount of reworked sediment input from continental shelf regions as a result of sea level drop. In the Antarctic Ocean, Ross Sea and Amundsen Sea (+West Antarctic Peninsula) also showed ice sheet dynamics associated OC burial during glacial periods (high OC, C/N in dark laminated mud with higher sedimentation rates than this study). Thus, the clear peaks of OC accompanying with distinct other proxies in the Arctic Ocean during glacial periods may not be a significant phenomenon in a global point of view.

Authors: Thanks for this important comment. We agree that although the glacial OC burial rates at our core site (i.e., the central Arctic Ocean) are 3-6 times higher than those determined for interglacial time intervals, these increased burial rates are still low in comparison to data from other continental margin areas. Nevertheless, thick turbidite deposits were deposited in glacial sediments of the central Arctic Ocean basins that are significant for budget calculations (although this is quite challenging due to missing age control (see line 411-422). Because most of the OC that accumulated in the Arctic deep-sea basins, is predominantly terrestrial (and ancient) OC, we weaken our too strong statement about the significance of the increased OC burial in central Arctic Ocean deep-sea sediments for the global OC and CO₂ budget (and by this for the Earth's climate system). This aspect, however, was not the main focus of our paper.

Reviewer 3: *Suggestions for additional references.*

Bollen et al. 2022. Pleistocene oceanographic variability in the Ross Sea: A multiproxy approach to age model development and paleoenvironmental analyses. Global and Planetary Changes 216.

Ha et al. 2022. Glaciomarine sediment deposition on the continental slope and rise of the central Ross Sea since the Last Glacial Maximum. Marine Geology 445.

Hillenbrand et al., 2021. New insights from multi-proxy data from the West Antarctic continental rise: implications for dating and interpreting Late Quaternary palaeoenvironmental records. 257.

Authors: Thanks for suggestions for further references. We have referred to Bollen et al. (2022) and Ha et al. (2022) in our discussion (line 396-402).

Bremen, 05.03.2025

Prof. Dr. Ruediger Stein
Faculty of Geosciences
University of Bremen
Klagenfurter Str. 2-4
D-28359 Bremen, Germany

Subj.: **Nature Communications manuscript NCOMMS-24-59102A**

Point-by-point response to the referees' comments

Following the suggestions/comments given by reviewers 2 and 4, we have carried-out a final revision of the last version of the manuscript. All our changes/corrections done in the manuscript text are highlighted by yellow background color.

(1) Comments of Reviewer #2 (in italics) and our/Authors' response

Reviewer 2: *"I have already reviewed this manuscript at the first round, and I can confirm this is a very interesting (thought impressive) work that identified key sites for stratigraphic, paleoenographic and paleoclimatic reconstructions in the central Arctic but also dealt with OC burial at high-latitudes in deep-sea sediments during glacial stages/glacial terminations having a strong impact on the global OC and CO₂ budget in Earth's climate system. If I may, I would like to suggest a couple of articles in support of their discussion: Lucchi et al., 2013 that reports about high concentration of OC buried in glacial diamicton on the western margin of Svalbard (Arctic) and Lucchi et al., 2007 that defined the concept of 'glacial contourites'....."*

Authors: We have included the references Lucchi et al. (2013) and Lucchi & Rebesco (2007) in our discussion.

Reviewer 2: *".....I made some additional, very minor, suggestions to the reviewed manuscript that are reported in the attached pdf file."*

Authors: We have considered all the (very minor) suggestions/corrections made by Reviewer 2 in lines 99-100, 133, 188, 247, 388, 393, 396, 419 and 426.

Reviewer 2: *".....The authors made a meticulous revision of the manuscript taking in consideration all the suggestion..... I consider the manuscript ready for publishing."*

Authors: Thank you very much for this very positive final statement.

(2) Comments of Reviewer #4 (in italics) and our/Authors' response

Reviewer 4: *"I have been asked to give thoughts on the authors' responses to reviewer #3's comments (reviewer #3 unavailable for the task), and was also welcomed to comment on the authors' responses to reviewer #2's comments, and to comment on the paper in the general as well, if I would like".*

Authors: Reviewer 4 went through Reviewer#3's comments and our responses very carefully, and was fully in line with all our responses. Furthermore, Reviewer 4 also found our responses/actions to reviewer #2's comments totally adequate. Thank you very much for these very positive statements! Thus, we have nothing to add here.

Reviewer 4: *"My overall evaluation is to recommend publication of this very nice paper."*

Authors: Thank you very much for this final statement!

The authors found that terrOC deposition increased during glacial (stadial) periods in the deep Eurasian Basin. These terrOC was transported by ice sheet advance ("pushed") across the shelf edge, then by mass-wasting processes to the down slope area. In addition, these OC is glacially eroded from subglacial sedimentary rocks and soils in the Taimyr-Severnaya Zemlya-Kara Sea area. Glacial OC burial rates are 3-6 times higher than interglacial times and the anoxic condition improved preservation of more labile algae-type OC. Therefore, they argued that Arctic deep-sea sediments were an important OC (and CO₂) sink during glacial periods. The proposal is interesting and the records show clear patterns; increased values in the dark layers and decreased values in the brown layers for all records. However, there are some issues should be solved.

1. chronology

Chronology is one of hot issues in the Arctic Ocean; lithostratigraphic chronology vs magnetic property based chronology (e.g., RPI and/or magnetic reversals). 5 pages of 16 pages of main text are about justification and how the age model was set. As a result, more attention is likely to be drawn to the chronological part rather than the key findings of this study. Why the proportion of chronological part is significant in this manuscript is related to the current age constraining issue. In this sense, I cautiously suggest that this manuscript might be more suitable for more specific journals rather than Nature Comm. which deals with broad interests rather than local/regional ones.

2. glacial erosion of subglacial sedimentary rocks and soils in the Taimyr-Severnaya Zemlya-Kara Sea areas?

High C/N ratios and high T_{max} support that OC is reworked and have terrestrial characteristics. Organic-geochemical and mineralogical proxies showed significant maxima in (Clino)pyroxene, smectite, kaolinite and siderite in high OC dark intervals originated from the above region. Subglacial sedimentary rocks and soils in that area are the source for high OC-rich dark layer. Sedimentary rocks mean (semi)consolidated form? After glacial erosion, chips or fragments of mudstone are expected to be found rather than mud form. However, very fine grain size are characteristics for the dark layers.

Ancient (patriotic) OC has lower T_{max} than relatively recently deposited brown layer sediments' OC? Is there any evidence for ancient OC? Such as microfossils, or some specific minerals found in the far past? In the Ross Sea, continental slope sediment study showed increased biogenic opal, TOC, and C/N (Ha et al., 2022) and a deep-basin sediment study also showed increased diatoms (reworked) in laminated mud during glacial periods (Bollen et al., 2022). They also interpreted that they are transported by turbidity current as ice sheet advanced across the continental shelf edge. But, Ha et al. (2022) suggested that the sediments deposited in the slope during glacial periods were originally

deposited on the shelf during former interglacials.

According to the authors, the dark layer sediments are a result of contour and turbidity currents (upward fining supports this). This means, they are laterally transported and likely sedimentation rates are high. However, marine and sea ice algae immediately increased as if they were already in the sediment. The strong correlation between TOC and all other proxies suggests that they were transported together rather than due to better preservation under anoxic conditions.

If authors want to argue that increases in marine and sea ice algae are due to surface biological activity under glacial polynas conditions, how much OC is derived from these glacial surface biological productions compared to terrOC should be calculated. This is important because authors also argued that the Arctic Ocean played a role as a sink for CO₂.

3. OC-3,-5,and -6 covers full glaciation to deglaciation phases. However, there is no difference between glacial part and deglacial part in the records although more subglacial melt water is likely expected during deglacial (retreating phase). How about outburst of meltwater and suspension changes according to ice sheet phases should be considered.

4. Although authors argued 3-6 times more OC burial occurred in the Arctic Ocean during glacial periods, 0.015-0.03g/cm²/kyr is not outstandingly high, but is relatively low compared to other regions. Thus, the increased OC burial during glacial periods are difficult to be thought as a major significance in the Earth's climate system.

More importantly during glacial periods continental margin deep sea areas receives huge amount of reworked sediment input from continental shelf regions as a result of sea level drop. In the Antarctic Ocean, Ross Sea and Amundsen Sea (+West Antarctic Peninsula) also showed ice sheet dynamics associated OC burial during glacial periods (high OC, C/N in dark laminated mud with higher sedimentation rates than this study). Thus, the clear peaks of OC accompanying with distinct other proxies in the Arctic Ocean during glacial periods may not be a significant phenomenon in a global point of view.

In summary, the clear cyclicity of lithology and OC, C/N, and all other biomarkers is interesting. However, why OC burial in the Arctic Ocean is important is not well convinced. In addition, its impact seems to be far more exaggerated, particularly global sink for OC and CO₂.

Bollen et al. 2022. Pleistocene oceanographic variability in the Ross Sea: A multiproxy approach to age model development and paleoenvironmental analyses. *Global and Planetary Changes* 216.

Ha et al. 2022. Glaciomarine sediment deposition on the continental slope and rise of the central Ross Sea since the Last Glacial Maximum. *Marine Geology* 445.

Hillenbrand et al., 2021. New insights from multi-proxy data from the West Antarctic continental rise: implications for dating and interpreting Late Quaternary palaeoenvironmental records. 257.

Comment on the paper "A 430 kyr record of ice-sheet dynamics, sediment discharge, and organic-carbon burial in the central Eurasian Arctic Ocean" (Nature Communications manuscript NCOMMS-24-59102A)

I have been asked to give my thoughts on the authors' responses to reviewer #3's comments (reviewer #3 unavailable for the task) on the above mentioned paper by Stein et al., submitted to Nature Communications, and was also welcomed to comment on the authors' responses to reviewer #2's comments and comment on the paper in the general as well, if I would like.

I have read the paper and my overall impression is that it is a very well written such, with a wealth of data giving insights into the glacial history for the European Ice Sheet in its eastern extreme over the past 430 kyr, as deduced from just one sediment core, just outside of the shelf break from the Kara Sea to the Arctic Basin. I am especially delighted with the author's concern to link the marine data to the terrestrial field data (as being a "land" geologist" myself), which is so often NOT done in marine-based works. And finding such a nice fit with that data (Taimyr Peninsula) at least for MIS 2 to MIS 6.

Reading through reviewer #3 comments to the original manuscript (which I have not read), and how the authors have responded to those comments, I give my comments to authors responses to those comments below.

Reviewer 3: *Chronology is one of hot issues in the Arctic Ocean; lithostratigraphic chronology vs. magnetic property-based chronology (e.g., RPI and/or magnetic reversals). 5 pages of 16 pages of main text are about justification and how the age model was set. As a result, more attention is likely to be drawn to the chronological part rather than the key findings of this study. Why the proportion of chronological part is significant in this manuscript is related to the current age constraining issue. In this sense, I cautiously suggest that this manuscript might be more suitable for more specific journals rather than Nature Comm. which deals with broad interests rather than local/regional ones.*

My comment: First, I would like to not agree with reviewer #3 on "I cautiously suggest that this manuscript might be more suitable for more specific journals rather than Nature Comm.". I find the paper highly suited for a broader public, as it is "one-of-a-kind" marine core that they have retrieved. I think authors response to the reviewer says this well.

Reviewer 3: *Glacial erosion of subglacial sedimentary rocks and soils in the Taimyr-Severnaya Zemlya-Kara Sea areas. Subglacial sedimentary rocks and soils in that area are the source for high OC-rich dark layer. Sedimentary rocks mean (semi)consolidated form? After glacial erosion, chips or fragments of mudstone are expected to be found rather than mud form. However, very fine grain sizes are characteristics for the dark layers.*

My comment: Find the authors give a solid explanation to reviewers' concern.

Reviewer 3: *Ancient (patriotic) OC has lower Tmax than relatively recently deposited brown layer sediments' OC?*

My comment: Authors have replotted misleading values in figure and solved the question.

Reviewer 3: *Is there any evidence for ancient OC? Such as microfossils, or some specific minerals found in the far past? In the Ross Sea, continental slope sediment study showed increased biogenic opal, TOC, and C/N (Ha et al., 2022) and a deep-basin sediment study also showed increased diatoms (reworked) in laminated mud during glacial periods (Bollen et al., 2022). They also interpreted that they are transported by turbidity current as ice sheet advanced across the continental shelf edge. But,*

Ha et al. (2022) suggested that the sediments deposited in the slope during glacial periods were originally deposited on the shelf during former interglacials.

My comment: The authors have expanded on this question in the revised manuscript in a satisfactorily way.

Reviewer 3: *According to the authors, the dark layer sediments are a result of contour and turbidity currents (upward fining supports this). This means, they are laterally transported and likely sedimentation rates are high. However, marine and sea ice algae immediately increased as if they were already in the sediment. The strong correlation between TOC and all other proxies suggests that they were transported together rather than due to better preservation under anoxic conditions.*

My comment: The authors give a long explanation why they do not agree with the reviewers concerns about the occurrence of marine algae abundance. I find that the authors' rebuttal on this issue is quite convincing. They have highlighted the issue in the revised manuscript.

Reviewer 3: *If authors want to argue that increases in marine and sea ice algae are due to surface biological activity under glacial polynyas conditions, how much OC is derived from these glacial surface biological productions compared to terrOC should be calculated. This is important because authors also argued that the Arctic Ocean played a role as a sink for CO₂.*

My comment: Authors conclude that such %-age division between the two sources is difficult to calculate, but explain that their studies suggest less than 10% of the OC is derived from marine/sea ice algae, which now is shown.

Reviewer 3: *OC-3,-5, and -6 covers full glaciation to deglaciation phases. However, there is no difference between glacial part and deglacial part in the records although more subglacial melt water is likely expected during deglacial (retreating phase). How about outburst of meltwater and suspension changes according to ice sheet phases should be considered.*

My comment: The authors have added text on this issue in revised manuscript.

Reviewer 3: *Although authors argued 3-6 times more OC burial occurred in the Arctic Ocean during glacial periods, 0.015-0.03g/cm²/kyr is not outstandingly high, but is relatively low compared to other regions. Thus, the increased OC burials during glacial periods are difficult to be thought as a major significance in the Earth's climate system. More importantly during glacial periods continental margin deep sea areas receives huge amount of reworked sediment input from continental shelf regions as a result of sea level drop. In the Antarctic Ocean, Ross Sea and Amundsen Sea (+West Antarctic Peninsula) also showed ice sheet dynamics associated OC burial during glacial periods (high OC, C/N in dark laminated mud with higher sedimentation rates than this study). Thus, the clear peaks of OC accompanying with distinct other proxies in the Arctic Ocean during glacial periods may not be a significant phenomenon in a global point of view.*

My comment: The authors has addressed this issue by weakening their before to strong statements on the significance in the revised manuscript, which I think is OK.

In addition, I have found a few minor errors in the manuscript:

Reference #8; publication date should be 2019 (not 2029)

Supplementary Fig. 2. A mismatch between lettering of figure frames and caption text. Text for frame (c) in caption text refers to frame (b) in figure. Text for frame (b) in caption text refers to frame (c) in figure.

I find the authors responses/actions to reviewer #2 comments totally adequate.

My overall evaluation is to recommend publication of this very nice paper.